# The Arctic sea ice extent change connected to Pacific decadal variability

Xiao-Yi Yang[1,2] ,Guihua Wang[3,4],Noel Keenlyside[5,6]

[1]State Key Laboratory of Marine Environmental Science, and College of Ocean and Earth Sciences, Xiamen University, Xiamen China.

[2] Southern Marine Science and Engineering Guangdong Laboratory (Zhuhai), Zhuhai China

[3] Department of Atmospheric and Oceanic Sciences &Institute of Atmospheric Sciences, Fudan University, Shanghai China

[4] CMA-FDU Joint Laboratory of Marine Meteorology, Shanghai China

[5] Geophysical Institute, University of Bergen and Bjerknes Centre, Bergen Norway

[6] Nansen Environmental and Remote Sensing Center, Bergen Norway

*Correspondence to*: Xiao-Yi Yang (xyyang@xmu.edu.cn)

**Abstract.** After an unprecedented retreat, the total Arctic sea ice cover for the post-2007 period is characterized by low extent and a remarkable increase in annual cycle amplitude. We have identified the leading role of spring Bering Sea ice in explaining the changes in the amplitude of the annual cycle of total Arctic sea ice. In particular, these changes are related to the recent occurrence of multi-year variability in spring Bering Sea ice extent. This is due to the phase-locking of the North Pacific Gyre Oscillation (NPGO) and the Pacific Decadal Oscillation (PDO) after about 2007, with a correlation coefficient reaching -0.6. Furthermore, there emerge notable changes in the sea level pressure and sea surface temperature patterns associated with the NPGO in the recent decade. After 2007, the NPGO is related to a quadrupole SLP anomalies that is associated with the wind stress curl and Ekman pumping rate anomalies in the Bering deep basin; these account for the change in Bering Sea subsurface variability that contribute to the decadal oscillation of the spring Bering Sea ice extent.

## 1 Introduction

The Arctic sea ice is an integral part of the Arctic climate system as well as one of the strongest indicators of global climate change. Since the satellite era, Arctic sea ice has declined along with the Arctic amplification of global warming. It is noteworthy that the remarkable ice retreat from the late 1990s (Comiso et al., 2008; Comiso, 2012; Stroeve et al., 2012) and the associated Arctic surface warming (Screen and Simonds 2010) is in great contrast to the simultaneous slowdown of surface warming in the Northern Hemisphere extra-polar regions (Cohen et al., 2014). Therefore, the possibility of "tipping points" or abrupt changes in the Arctic climate system has come to be a subject of much debate (Lindsay and Zhang, 2005; Lenton et al., 2008; Eisenman and Wettlaufer, 2009; Armour et al., 2011; Serreze, 2011; Tietsche et al., 2011; Duarte et al.,

2012; Bathiany et al., 2016a). There are two aspects to the controversy: 1. Whether or not the Arctic sea ice cover will melt persistently and irreversibly to an ice-free state; 2. Whether or not a bifurcation with multiple steady states exists such that the Arctic may transition from perennial to seasonal ice cover. In general, model studies do not support the existence of threshold or "tipping point" behaviour. By prescribing atmospheric carbon dioxide, Armour et al. (2011) and Bathiany et al.

(2016a) demonstrated that there is neither irreversibility nor multiple states for the total Arctic sea ice cover. Tietsche et al. (2011) further proposed a recovery mechanism for Arctic summer sea ice based on the anomalous long-wave emission and atmospheric heat advection. Nevertheless, statistical analysis of observations presents various early warning signals of abrupt Arctic climate change. For example, Duarte et al. (2012) argued that it is the ice thickness rather than ice extent that cannot be rebuilt. With the persistent thinning of Arctic sea ice, open-water formation increase during the melting season is

probably followed with higher efficiency of ice production during the next ice growth season through the ice thickness-ice growth feedback (Barnhart et al., 2016). Massonnet et al. (2018) further suggested that the reductions in ice thickness may lead to enhanced seasonality but decreased interannual variability and persistence of the Arctic sea ice extent. Livina and Lenton (2013), by the "potential analysis", detected the abrupt and persistent increase in the seasonal cycle of Arctic sea ice cover since 2007. Therefore, the impact of climate change on Arctic sea ice is much more pronounced in the amplitude of

the seasonal cycle than in the annual mean ice area.

While previous studies mostly considered the signal of abrupt Arctic climate change in terms of rapid sea ice decline and the amplifying seasonal cycle, few of them focused on the regional and seasonal dependence of Arctic sea ice variations. Close et al. (2015) concluded that the onset of Arctic sea ice decline is much earlier, and the post-onset rate of loss is weaker in the Pacific than in the Atlantic sector. Onarheim et al. (2018) noted the distinct transformation of seasonal sea ice

variability in various Arctic regional seas and predicted the increase and expansion of ice-free periods into the Arctic in the future. These results indicate that Arctic sea ice variations are asymmetrical and strongly dependent on the local physical process, resulting in the complexity and difficulty in predicting the future Arctic climate change. Therefore, we need to probe into the recent Arctic sea ice variability, both temporally and spatially. It is urgent to understand the physical mechanisms accounting for the recent amplifying seasonal cycle of Arctic sea ice cover, so as to better foresee future Arctic

climate change.

## 2 Data and methods

We used the monthly sea ice concentration (SIC) dataset that was derived using the NASA team algorithm from the *Nimbus*-7 SMMR (1978-87), DMSP SSM/I (1987-2009), and DMSP SSMIS (2008-present) satellite passive microwave

radiances on a 25km×25km polar stereographic grid (Cavalieri et al., 2013); the dataset can be freely downloaded from National Snow and Ice Data Center (NSIDC) website. The sea ice extent (SIE) for the period from November 1978 to June 2019 was generated by summing the areas of the grid boxes covered with at least 15% ice concentration for each month. Note that this timeseries has negligible differences to the one computed from daily SIC with the same threshold and then averaging over the month.

We also used surface air temperature, sea level pressure, 850hPa zonal and meridional winds, and sea surface temperature from the European Centre for Medium-Range Weather Forecasts (ECMWF) ERA-interim reanalysis dataset (Dee et al., 2011), with a horizontal resolution of $0.75° \times 0.75°$ and ranging from January 1979 to December 2018.

The ORAS4 ocean temperature, salinity and horizontal velocity data were used to examine ocean subsurface variability. The ORAS4 is a newly-released version of ECMWF ocean reanalysis dataset, which is based on the NEMO ocean model and assimilates the available temperature and salinity profiles and sea-level anomalies through the NEMOVAR assimilation scheme (Balmaseda et al., 2012). The dataset spans the period of January 1958 to December 2017, with the resolution of $1° \times 1°$ grid horizontally and 42 levels vertically.

We used two monthly climate indices: The Pacific Decadal Oscillation (PDO) index, downloaded from the Joint Institute for the Study of Atmosphere and Ocean (JISAO) website at http://research.jisao.washington.edu/pdo/; and the North Pacific Gyre Oscillation (NPGO) index, constructed by Emanuele Di Lorenzo (Di Lorenzo et al., 2008) and downloaded from http://www.o3d.org/npgo/.

The National Centers for Environmental Prediction (NCEP) Global Ocean Data Assimilation System (GODAS) data, HadISST1 sea surface temperature data and the Hadley Centre EN4 quality-controlled ocean data were also used to validate the results.

The surface turbulent heat flux from the Woods Hole Oceanographic Institution (WHOI) Objectively Analyzed Air-Sea Fluxes (OAflux) dataset from January 1958 to December 2018 was also used in this study. The spatial resolution of the data is $1° \times 1°$. The OAflux dataset is constructed from multiple data sources, with an optimal blending of satellite retrievals and reanalyses data (Yu et al., 2008).

The amplitude of seasonal cycle is estimated by calculating the annual standard deviation based on the following formula:

$$\text{std}_j = \sqrt{\frac{1}{12} \sum_{i=1}^{12} (x_{ij} - \overline{x}_j)^2}$$

where the subscripts of i, j are the month and the year, respectively. The overbar denotes the annual mean for the $j^{th}$ year.

The multi-year averaged monthly standard deviation is computed as:

$$\text{std}_i = \sqrt{\frac{1}{n} \sum_{j=1}^{n} (x_{ij} - \overline{x}_j)^2}$$

where n denotes the number of years for averaging.

A harmonic analysis method based on fast Fourier transform is used to extract the decadal component of the spring Bering sea ice index, and the PDO and NPGO indices.

# 3 Results

## 3.1 Decadal change of Arctic sea ice extent variance

The time series of monthly total Arctic SIE anomalies from November 1978 to June 2019 is shown in Fig. 1a (upper panel), together with the magnitude of the annual standard deviation (lower panel in Fig. 1a). The total Arctic sea ice cover variability during the satellite period goes through three distinct stages: relatively high and stable sea ice cover during 1979-1998; fast ice retreat from 1999 to 2006; and the lower ice extent and level-off in the ice retreat trend post-2007. During the 1999-2006 period sea ice declines with a dramatic rate of over -0.13 million $km^2$/year, such that the climatology of total Arctic sea ice extent drops from ~13 million $km^2$ for the 1979-1998 period to ~11.6 million $km^2$ for the post-2007 period. In association with the slow-down of Arctic sea ice retreat, there is an abrupt increase in the magnitude of annual ice fluctuations in the recent decade. In particular, the mean value of annual standard deviation leaps from 3.26 million $km^2$ before 2007 up to 3.79 million $km^2$ in the period of 2007-2012. Thereafter the mean annual standard deviation drops to 3.54 million $km^2$ in 2013-2018. Therefore, the current state of Arctic sea ice cover is characterized by lower coverage, a level-off in the linear decline trend, and an intensification of the seasonal cycle. All these signals herald the transition to another Arctic climate regime, which is a hot topic in the climate research community (Livina and Lenton 2013; Bathiany et al., 2016b).

In view of the strong changes in Arctic sea ice seasonality, we further examine the decadal changes of sea ice extent between the three periods (i.e., 1979-2006, 2007-2012 and 2013-2018) for different calendar months (Fig. 1b). Following the definition of multi-year averaged standard deviation (Section 2), monthly deviations from the annual mean are squared and averaged for the separate three periods. The square roots are taken and the differences between the first two periods (black line) and the last two periods (red line) are estimated for each month. Both the increase in 2007-2012 and the decrease afterwards in the annual standard deviation are associated with the same bimodal change in standard deviation for different calendar months, with the main peak in September and the secondary peak in April-May. The spatial distribution of changes in the standard deviation of sea ice concentration is presented by calculating the sum of the absolute differences in the standard deviation in these two peak seasons (summer August-October mean and spring March-May mean; fig. 1c & d). There is a great contrast between the Pacific and the Atlantic sectors in both seasons: while the Pacific sector including the Bering Sea, the Chukchi Sea, the East Siberian Sea, the Laptev Sea and the Beaufort Sea contributes to the observed decadal changes of total Arctic sea ice extent (fig. 1a & b), the Atlantic sector including the Barents Sea, the Greenland Sea and the Baffin Bay offsets these changes. It is notable that the maximum variance changes are located in the north and south sides of the Bering Strait, i.e., the Chukchi Sea in summer and the Bering Sea in spring (labeled the ChukBer region, and demarked by the grey fan-shaped frame), following the seasonal advance and retreat of climatological sea ice extent. In addition, the climatological mean sea ice extent for three periods also exhibit decadal changes, with the 2007-2012 mean ice edge retreating to the northernmost in summer (fig. 1c green line) and extending to the southernmost in spring (fig. 1d green line); this is consistent with changes in annual standard deviation (fig 1a, bottom). Thus, the ChukBer region plays a key role in the

recent change in Arctic sea ice seasonality and variability, though the sea ice variations in the Chukchi Sea and the Bering Sea are probably subject to different dynamical processes and thermal conditions.

Fig. 2 shows the normalized ChukBer sea ice extent indices in summer and spring seasons. In summer the index represents variations in sea ice extent mainly in the Chukchi Sea, because of the seasonal retreat of sea ice; while in winter it represents variations in the Bering Sea. The summer Chukchi sea ice variability exhibited a tendency similar to the total Arctic sea ice variability: relative stable sea ice cover in early period, followed by a fast decline from the late 1990s to the early 21$^{st}$ century. After 2007, the Chukchi summer sea ice cover has remained in a low extent and leveling-off stage in the recent decade. The Bering Sea ice in spring, however, varied in a quite different pace. The apparent transition from a moderate interannual variability to decadal-scale variability with much larger magnitude emerged around 2007. The contrasting behavior is consistent with the changes annual standard deviation: anomalous high ice extension in spring and lower ice coverage in summer in the ChukBer region result in the abrupt increase of seasonal cycle magnitude for the period of 2007-2012; thereafter, the spring Bering ice cover retreated remarkably, accounting for the retracement of total Arctic sea ice annual standard deviation for the period 2013-2018.

It is widely recognized that significant summertime ice loss in the Chukchi Sea is primarily attributed to the (thermodynamic) ice-albedo positive feedback and (dynamic) strengthening transpolar drift. So far there is a lack of comprehensive understanding and interpretation for the recent decadal change of Bering ice extent in spring. Therefore, a composite analysis is applied to various oceanic and atmospheric fields to explore the physical mechanism that triggered the decadal change of spring Bering Sea ice extent. Based on the threshold of $\pm 0.7$ standard deviation of spring Bering sea ice extent index (SBII), a total of 18 out of 40 years are selected to do the composite analysis, including 10 high index years (1984, 1992, 1994, 1995, 1999, 2008, 2009, 2010, 2012 and 2013) and 8 low index years (1979, 1989, 1996, 2003, 2015, 2016, 2017, 2018).

### 3.2 Abrupt transition of air-sea interaction in the Pacific sector

As the sea ice is sensitive to local thermal conditions in the ocean-atmosphere interface, the March-May mean surface air temperature, sea level pressure and 850hPa wind anomalies are composited based on the SBII index (high index years minus low index years) for two separate periods of 1979-2006 (Fig. 3 left panels) and 2007-2018 (Fig. 3 right panels). For both periods, the positive phase of SBII corresponds to the significant cooling (negative anomalies of surface air temperature) over the Bering area (Fig. 3a & b). However, the SLP and 850hPa wind anomalies associated with the SBII present completely different patterns. In 1979-2006, the Chukchi-Bering region is dominated by an anomalous SLP dipole pattern and attendant northerly winds in the lower troposphere (Fig. 3c). The associated cold advection in lower troposphere is consistent with the drop in local air temperature (Fig. 3a). In contrast to the early period, the spring ice expansion after 2007 is associated with a significant high SLP anomaly over the Bering Sea and patchy northerly winds on its eastern flank (Fig. 3d), and these can hardly account for the broad cooling over the Bering region and the adjacent northeastern Pacific (Fig. 3b).

The SBII-related March-May mean sea surface temperature and subsurface (165m) temperature anomalies are shown in Fig. 4. For the early period, the positive SBII phase is associated with the local surface cooling that is confined to the Bering Sea area (Fig. 4a). In the later period, however, the spring Bering Sea ice expansion is connected to a large-scale SST anomaly pattern: the significant cooling in the Bering Sea extends along the North America coast to the central tropical Pacific, sandwiching the warming anomalies in the adjacent area of the Kuroshio extension (Fig. 4b). In addition, cooling anomalies in the early period are mostly limited to the sea surface, while those in the later period extend to the subsurface layer (Fig. 4d).

To further assess the connection between the post-2007 SBII decadal change and local thermal conditions, the surface air temperature, sea surface temperature and subsurface temperature anomalies from various datasets are averaged in the Bering region (170-200°E, 55-65°N, delimited by the grey rectangles shown in Fig. 4) and low-pass filtered by a 13-month running mean. All the time series during the period of 1950-2018 are plotted in Fig. 5. The Bering mean air temperature shows moderate negative anomalies in 2007-2012 but remarkable warming afterwards (Fig. 5a). Although there are subtle differences in terms of magnitude and timing, the persistent cooling during 2007-2012 and warming in recent five years are evident in all the SST and subsurface temperature data (Fig. 5b-f). This indicates that the recent SBII change may connect more to the large-scale oceanic dynamical processes or ocean-atmosphere coupled modes rather than to the local atmospheric circulation. Here we used all-month data instead of March-May mean because the seasonality in subsurface temperature is very weak, and the post-2007 changes of subsurface temperature are prominent in all seasons. For the surface temperature, however, the seasonality is strong: the SST decadal oscillation can be detected in the winter and spring seasons, but obscured by a long-term warming trend in the summertime; the surface air temperature shows even weaker signals of decadal change in the cold season and stronger warming trends in the warm season (figures not shown).

### 3.3 NPGO pattern change and its synchronization with the PDO

Numerous previous studies asserted that extra-tropical North Pacific climate variability is dominated by two decadal-scale ocean-atmosphere coupled modes: the Pacific Decadal Oscillation (PDO) (Mantua et al. 1997) and the North Pacific Gyre Oscillation (NPGO) (Di Lorenzo et al. 2008). Though defined independently as the first and second basin-scale EOF modes of sea surface temperature anomalies and sea surface height anomalies in the north Pacific Ocean respectively, the PDO and NPGO bear strong resemblance in their physical mechanisms and related climate anomalies. They are both forced by the El Niño-like tropical Pacific SSTA (Newman et al. 2003; Di Lorenzo et al. 2010), connected to the western Pacific Kuroshio-Oyashio extension through the ocean Rossby wave propagation (Schneider and Corneulle 2005; Ceballos et al. 2009), and impact the low-frequency variability of the marine ecosystem system (Miller et al. 2004; Di Lorenzo et al. 2013; Sydeman et al. 2013). Besides the remote tropical ENSO forcing, the PDO and NPGO can be driven by the local ocean-atmosphere interaction. The ocean, as a low-frequency filter, integrates the atmospheric stochastic noise and thereby contributing to the decadal-scale variability (Newman et al. 2016; Yi et al. 2015). The corresponding large-scale atmospheric modes are the Aleutian Low variability (for the PDO) and the North Pacific Oscillation (for the NPGO).

The large-scale Pacific SST anomaly pattern in the later period (Fig. 4b) alludes to the PDO and the NPGO as the candidates responsible for the decadal variations in the SBII during the most recent period. Therefore, changes in the spring (March-May mean) PDO and NPGO patterns are assessed by regressing the SST and SLP anomalies onto the two indices respectively for the different periods (Fig. 6). The PDO positive phase is associated with the deepening of the Aleutian Low,

warming in the northeast Pacific, including in the Gulf of Alaska and along the California coast, while there is cooling in the Kuroshio extension. Moderate warming in the eastern equatorial Pacific is discernable (Fig 6a). In the recent decade, the PDO-related SLP pattern is almost unchanged, but the eastern Pacific warming enhances greatly. Notably the Bering Sea and the central equatorial Pacific stand out to be among the most significant warming regions (Fig 6c). The spring positive NPGO corresponds to an anomalous high pressure over most of the North Pacific and negative SLP anomalies over the

eastern Asian region and Alaska. The SSTA spatial pattern looks like the PDO SSTA pattern shown in Fig. 6a, but there are differences in spatial structure in the extratropical Pacific (Fig. 6b) that are consistent with a degree of orthogonality between these two modes (although defined on different quantities). The change of NPGO pattern after 2007 is conspicuous: There is a quadrupole SLP anomaly pattern that is associated with a significant cooling that extends all the way from the Bering Sea, to the northeastern Pacific, and to the central tropical Pacific (Fig. 6d); furthermore the NPGO pattern during this period

shows similarities to the PDO pattern. Thus, both the PDO and the NPGO may collaborate in driving the Bering SST anomalies in the later period.

To investigate the time evolution of the PDO and NPGO impact onto the Bering Sea, we regress subsurface ocean temperature averaged over the Bering Sea onto the PDO and NPGO indices using a 241-month running window over the period of 1950-2018 (Fig. 7a & b). The ocean temperature in the Bering Sea indeed exhibits close connection with the PDO

since the mid-1970s; a weak connection prior to this period may reflect sparseness of subsurface observations. The correlation first appears in the subsurface layer, then enhances and extends upward with the time, reaching its peak near the surface after 2000. Positive one standard deviation of the PDO index corresponds approximately to a maximal 0.35℃ warming in the Bering Sea surface layer. The impact of NPGO, however, on the Bering Sea is significant only during the 1970s-1980s and after 2005. In addition, post-2005 the cooling associated with the NPGO positive phase is greater in the

subsurface layer rather than at the surface. We further calculate the partial correlation coefficients between the Bering temperature and the PDO (NPGO) with the effect of NPGO (PDO) removed (Fig. S1). The spatial patterns of partial correlation analysis differ little from those in Fig. 7a & b; this corroborates the robustness of the different PDO and NPGO relation to the surface and subsurface ocean temperature in the Bering Sea.

The PDO and NPGO indices become more correlated during the most recent period, as shown by a 241-month running

correlation between the two indices (Fig 7c). The PDO and NPGO remain almost uncorrelated until the end of 1980s, thereafter they become more and more correlated. In the 21[st] century, the monthly correlation coefficient reaches -0.4, and even reaches -0.6 when seasonal variations are filtered out. The increasing synchronization of the PDO and NPGO modes can be clearly detected by comparing their time series (Fig. 8). The decadal amplitude of NPGO is almost doubled from the

end of 1980s. The strengthening of NPGO coalesces with the PDO to be in an apparent anti-resonance after 2007 (as the patterns are negatively spatially correlated, this SST patterns reinforce each other).

Yeo et al. (2014) also demonstrated that the Chukchi-Bering climate variability for the post-1999 period is closely related to Pacific basin-scale atmospheric and oceanic circulation patterns (i.e., the North Pacific Oscillation (NPO) and the NPGO), as well as the central Pacific El Niño. Indeed, the SBII time series in Fig. 2b exhibits an obvious change in its periodicity. The SBII varied mainly on interannual time scales before 2000, and thereafter on decadal time scales, coinciding with the NPO-NPGO variability in 1999-2010. But it is only after ~2007 when the NPGO and PDO indices vary out-of-phase that there is an obvious decadal change in the Arctic sea ice variance and a decadal oscillation in the SBII. The detailed physical mechanism for the apparent changes in the PDO and NPGO patterns deserves further investigation that is beyond the scope of this paper.

### 3.4 Distinct effects of PDO and NPGO on the Bering Sea

From the statistical analyses in the section 3.3, the NPGO and PDO changes may be responsible for the recent increase in SBII multi-year variability via their influence on the surface and subsurface ocean temperature in the Bering Sea. Here we investigate the physical linkages between these large-scale climate modes and the Bering Sea. In spite of high resemblance of their SSTA pattern after 2007, the SLP patterns related to the PDO and the NPGO differ greatly (Fig. 6 c & d). This implies distinct oceanic mechanisms in these two modes. The atmospheric thermal forcing is estimated by regressing the turbulent heat flux anomalies onto the PDO and the NPGO indices (Fig. 9). There are hardly any significant PDO-related heat flux anomalies in the Bering Sea region during either period (Fig. 9a & c). The positive phase of NPGO corresponds to the negative anomalies of heat flux along the northwest coast of Bering in the early period (Fig. 9b). In the later period, the NPGO is associated with positive and negative anomalies of heat flux appear in the southern and northern Bering Sea respectively, with the zero correlation line approximately along the climatological spring Bering Sea ice edge (Fig. 9d). Referring to the NPGO-related quadruple SLP pattern (Fig. 6d), the anomalous atmospheric cold advection associated with the anomalous high pressure prevails over the Bering Sea, and is consistent with the drop in atmospheric surface temperature and increased heat loss over the open ocean, and hence the SST cooling. The sea ice expansion follows the decreasing temperature and insulates the ocean-atmosphere exchange, resulting in the negative anomalies of heat flux over the ice cover. This may partly explain the enhanced Bering Sea surface cooling associated with the NPGO in the later period (Fig. 6d). It should be noted that the NPGO-related high-pressure anomalies over the Bering Sea extends westward to east Siberia, which is quite similar to the SBII-related SLP anomalous structure (Fig. 3d). This underlines the contribution of NPGO atmospheric anomalies in forcing the Bering Sea surface response.

The atmospheric dynamical forcing on the upper ocean circulation is assessed by regression of spring wind stress and Ekman pumping rate anomalies onto the PDO and NPGO indices (Fig. 10). In response to the deepening of Aleutian Low (Fig. 6a & c), positive phase of PDO is associated with an anomalous cyclonic circulation over the north Pacific, without any notable change between the two periods considered. The relevant positive Ekman pumping anomalies mainly reside in the

north Pacific open ocean, while the enhanced northeasterly prevails over the entire Bering Sea (Fig. 10a & c). In contrast, the NPGO has hardly any impact on the large-scale circulation. Nevertheless, the scattered significant wind stress anomalies in the later period organize an anomalous anti-cyclonic pattern, hence the negative Ekman pumping rate (downwelling) in the western Bering Sea (Fig. 10d). The vertical velocity induced by surface wind stress curl usually results in the displacement of thermocline or pycnocline, followed by the dynamical adjustment of the subsurface ocean.

The climatological upper ocean circulation in the Bering Sea and the adjacent region is primarily characterized by a cyclonic circulation, primarily composed of warm Alaska stream, cold Kamchatka current, and the Bering slope current (Stabeno et al., 1999). Given the significant wind stress forcing in accordance with PDO, the spring ocean velocity and density anomalies are regressed on the PDO index both in the surface (5m) and subsurface (165m) layer (Fig. 11). The regression is done for the whole period because the PDO exhibits little change between the periods. The large-scale cyclonic wind stress anomalies associated with the PDO positive phase lead to the strengthening of Alaska stream and enhanced northward heat transport along the Bering slope, contributing to the surface warming and density decrease in the Bering Sea (Fig. 11a). In the subsurface layer, however, the wind-driven component abates rapidly with depth. The anomalous heat transport is limited to the boundary current around the southern Bering deep basin (Fig. 11b). Therefore, the Bering Sea warming associated with PDO peaks in the surface layer. The circulation anomalies corresponding to the NPGO is also investigated, but without any remarkable signal of current and heat transport anomalies (figure not shown).

In consideration of the significant NPGO-related Ekman pumping rate (Fig. 10d) and subsurface cooling (Fig. 7b), the pycnocline displacement and vertical exchange between the mixed layer and subsurface ocean in the Bering Sea deep basin (160-190°E, 50-60°N) is investigated. The climatological vertical profile of temperature (T), salinity (S) and density (Rho) in this region in spring is presented in Fig. 12. The vertical stratification of Bering basin depends on salinity, with the closely matching position of the pycnocline and halocline at about 100-300m depth. In striking contrast to the increase of salinity and density with depth, the temperature profile exhibits a sandwich-like pattern: The relative warm subsurface water with its temperature exceeding 3.5℃ is sandwiched by the colder surface and deep water layers. It is clear that the cold and fresh water in the mixed layer is superposed over the warm and saline water in the pycnocline. The spring Bering Sea deep-basin hydrographic anomalies associated with positive NPGO for different periods are shown in Fig. 13. In the early period, the NPGO-related changes of water property are confined to the mixed layer. In the later period, the hydrographic anomalies associated with positive NPGO are almost doubled in the mixed layer, and a change in the pycnocline can be clearly detected. The colder and saltier water in the surface overlies the less cold and fresher water in the subsurface, corresponding to the density increase (decrease) in the mixed layer (pycnocline).

A NPGO-related atmosphere-ocean-ice mechanism may be depicted as follows: The quadrupole SLP pattern in the recent decade (Fig. 6d) favors the atmospheric cold advection and anticyclonic wind stress curl (Fig. 10d) over the Bering Sea in association with positive phase of NPGO. In response to atmospheric thermal forcing, ocean heat loss enhances in the open water (Fig. 9d), leading to the greater surface cooling. On the other hand, the Bering sea ice cover expands with the cooling condition. Ice expansion is usually followed by surface salinification in the adjacent open water owing to the brine-

rejection effect. Surface water is getting denser with the anomalous cooling and salinification. Triggered by the anticyclonic wind stress curl over the Bering basin region, anomalous downwelling acts to push the subsurface isopycnals downward, hence the decrease of density and freshening in the pycnocline with the positive NPGO (Fig. 13b). The reverse processes may occur for the negative NPGO phase: atmospheric warm advection leads to less heat loss from ocean and retreat of sea ice cover, thus warming and freshening of surface water. The surface density decreases to a large extent. The cyclonic wind stress forcing and the subsequent upwelling induce the heaving of subsurface isopycnals, hence increased density and salinification in the pycnocline. The NPGO leads to variations over deeper part of the upper ocean and this can explain a stronger impact on the seasonal cycle, as compared to periods when only the PDO influences the upper ocean of the region.

To further verify the sensitivity of SBII to the NPGO-oriented physical processes in the most recent period, the spring wind stress and Ekman pumping rate anomalies are composited based on the SBII high minus low index years for both periods (Fig. 14). The spring Bering sea ice cover expansion corresponds to the prevailing northerly over the Bering Sea and a large-scale anticyclonic circulation and upwelling in the North Pacific Ocean in the early period (Fig. 14a), which resembles the PDO-related pattern (Fig. 10a). For the later period, there are anomalous anticyclonic wind stress curl over the Bering Sea and downwelling in association with the increased ice extent (Fig. 14b), and this resembles the NPGO-related pattern (Fig. 10d). In addition, the Bering deep-basin water property changes in association with the SBII high minus low index years are presented in Fig. 15. In the early period, the ice expansion corresponds to the surface cooling but little change in the salinity and density (Fig. 15a). In the later period, the ice expansion corresponds to much larger magnitude of surface cooling as well as salinity and density anomalies. The changes can also extend to pycnocline layer with the reverse of the salinity and density anomalies vertically, and they resemble the NPGO-related hydrographic anomalies (Fig. 13b). This corroborates that the recent decadal oscillation of the Bering sea ice is primarily attributed to the NPGO pattern change.

**4 Conclusions and discussion**

By dissecting spatial-temporal variability of the Arctic sea ice, this study identifies that the recent multi-year variability in spring Bering Sea ice extent is one of the main factors enhancing the seasonality of total Arctic sea ice. The physical mechanism underlying the decadal change of spring Bering Sea ice extent is explored. The results show that the prominent decadal oscillation of spring Bering Sea ice is triggered by the NPGO pattern change and its synchronization with the PDO. The PDO influences the SST anomalies in the Bering Sea via its effect on ocean heat advection, which shows little variation during the different periods. The NPGO, on the other hand, shows conspicuous changes: while before 2007 the NPGO has little relation to the Bering basin, after 2007 it is associated with an SLP quadrupole pattern with anomalous anticyclonic wind stress curl over the Bering basin. The subsequent Ekman pumping rate anomalies induce the vertical undulation of pycnocline, leading to significant water property change in the subsurface layer.

It is worth noting that the decadal oscillation of Bering sea ice extent occurs only in spring season, but not in winter. We then examine the time series of winter (Dec-Feb mean) Bering sea ice extent. The winter Bering sea ice cover remains relative stable until 2014. Thereafter a drastic ice decline occurs in the recent five years (Fig. S2), which is highly consistent

with the abnormal atmospheric warming over the Bering Sea (Fig. 5a). The regressions of various physical fields onto the PDO and NPGO are repeated for the winter (Dec-Feb mean) season (Fig. S3 & S4). The SST and SLP anomalies associated with the PDO in winter season (Fig. S3a & c) looks much like those in spring season. The NPGO-related anomalous SST pattern shows little changes in winter, comparing to the pattern in spring. But there is discernable discrepancy of SLP anomalies between the two seasons. The winter NPGO-related SLP anomalies for the prior-2007 period exhibits the typical North Pacific Oscillation (NPO) pattern, with the SLP negative anomalies and positive anomalies to the north and south side of 45°N, respectively (Fig. S3b). For the post-2007 period, the NPO-like SLP dipole pattern seems shifted southeastward and anomalous high pressure appears over east Siberia. The Bering Sea is located at the nodal line between the positive SLP anomaly at the northwest and negative SLP anomaly at the southeast (Fig. S3d). As a result, northeasterly wind prevails over the Bering Sea without any marked anomaly in wind stress curl and Ekman pumping rate (Fig. S4d); thus there is no subsurface response. Therefore, the winter Bering Sea ice varies along with the surface air temperature, but with no decadal oscillation. This further demonstrates that the subsurface ocean response to the NPGO-related surface atmospheric forcing is a key process for the Bering Sea ice decadal change in spring.

A natural question is why the NPGO pattern exhibits such a remarkable change in spring and its evolution coincides with PDO after 2007. As the second mode of sea surface height anomalies in northeast Pacific, the NPGO mode essentially reflects changes in the intensity of subtropical and subpolar gyre circulations. It is driven either by local stochastic atmospheric processes (represented of North Pacific Oscillation) or by teleconnections to remote tropical ocean forcing (mostly related to central-Pacific El-Niño event) (Di Lorenzo et al. 2008; 2010). Therefore, the NPGO changes may be related to changes in the background atmospheric circulation and in the tropical Pacific thermal state. The Pacific atmospheric circulation is dominated by the Aleutian Low (AL) variability and/or its coherence with the subtropical Hawaii High, and can be largely represented by a North Pacific Index (NPI, Hurrell et al., 2019). The low-frequency (20-year low-pass filtered) variability of the spring (March-May mean) NPI shifted from the negative phase (deepening of Aleutian Low) since 1979 to the positive phase around 2007. Correspondingly, the climatological spring SLP pattern post-2007 shows a weak eastward shift of Aleutian Low relative to the prior-2007 period (Fig. S5). This may contribute to the shift of NPGO-related SLP anomalies, from NPO-like dipole pattern to a quadrupole pattern. The SST pattern change in NPGO mode may also be related to changes in the tropics. Numerous studies have asserted the increasing ratio of central Pacific warming to the canonical eastern Pacific warming since the late twentieth century (Lee and McPhaden 2010; Yu et al. 2010; Liu et al. 2017; Freund et al. 2019). Thus, the transition of tropical Pacific thermal state may also have triggered the decadal changes of NPGO. We further examine the NPGO definition and repeat the same SSH EOF analysis in the Northeast Pacific (180°W-110°W, 25°N-62°N) as Di Lorenzo et al. (2008). The running correlation of Lorenzo's NPGO index and the SSH EOF PC1 and PC2 shows an obvious decadal change in the late 1990s, after which the NPGO is much more significantly correlated with the EOF PC1 than PC2 (Fig. S6). Further SSH EOF analysis during two separate period (1979-1998 vs. 1999-2017) confirms that spatial pattern of SSH second EOF mode in the later period is totally different from that in the early period, as well as from canonical NPGO pattern (Fig. S7). The explained variance of 2nd EOF of SSH decreased from about 14% in the

1980s and 1990s to 10% in the new century. At the same time, the connection between the NPGO and its atmospheric counterpart—the NPO also weakened in the recent decade. Both the oceanic and atmospheric representations of NPGO undergo strong decadal changes, which may be related to its synchronism with PDO. More detailed analysis is needed to explore the connection of tropical Pacific thermal state change with the northeast Pacific dynamical response.

The study of Zhang and Knutson (2013) revealed a great contrast between the rapid decline in the summer Arctic sea ice extent and the relatively "flat" trend in the global-mean surface air temperature in the early 21$^{st}$ century. Zhang (2015) identified the ocean heat transport from Pacific as one of the key predictors of the internal variability of summer Arctic sea ice extent. Woodgate et al. (2010) further reported an increase in oceanic heat flux through the Bering Strait since the early 2000s, and proposed that this Bering Strait inflow may act as a conduit for oceanic heat into the Arctic and contribute to the

unprecedented seasonal Arctic sea-ice loss since 2007. Echoing with this increase heat flux into the Arctic Ocean, the Bering Sea is reported to be the only substantial non-coastal area with lengthening sea ice seasons within the Pan-Arctic region before 2013 (Parkinson, 2014). However, the Bering Sea ice cover in these five years experiences a cascading decline, particularly in 2018 and 2019 years. This abrupt change may be attributed to the collective effect of Arctic climate change and the recent Pacific decadal variability, i.e., the Arctic surface warming and the local subsurface change. The Bering Sea

change, in turn, could feedback to the central Arctic Ocean by modulating Bering Strait throughflow strength as well as the heat and freshwater transport through the strait. The model simulation by Lee et al. (2013) demonstrated that in response to the climate warming from a doubling of $CO_2$, the maximum sea ice extent in the Bering Sea retreats to the north; this is associated with the increase of northward heat transport through the Bering Strait in early winter and spring. Therefore, further study is necessary to determine the complicated and interweaving physical processes that may lead to the recent

Arctic climate change. In the background of global warming and polar amplification, the total Arctic ice volume and thickness decrease persistently. It is conceivable that the future change of Arctic sea ice extent is more sensitive to the complicated air-sea coupling processes. Here we have contributed to further understand the Bering Sea thermal state and dynamical processes, as well as its connection to the North Pacific climate change. Further work is required in order to answer questions like whether this tendency of amplifying seasonal cycle of Arctic sea ice cover will be sustained and

whether it will lead to a nonlinear instability in the Arctic climate system.

*Data availability*. NSIDC monthly sea ice satellite product can be downloaded via http://nsidc.org/data/nsidc-0051.html and http://nsidc.org/data/nsidc-0081.html. ERA-interim reanalysis dataset is provided by ECMWF on their website https://apps.ecmwf.int/datasets/data/interim-full-moda/levtype=sfc/. ORA-S4 ocean reanalysis dataset is supplied by

https://climatedataguide.ucar.edu/climate-data/oras4-ecmwf-ocean-reanalysis-and-derived-ocean-heat-content. OAflux surface turbulent heat flux data is downloaded via ftp://ftp.whoi.edu/pub/science/oaflux/data_v3. GODAS reanalysis, HadISST and HadEN4 subsurface ocean data are available from Asia-Pacific data-research center of the IPRC via http://apdrc.soest.hawaii.edu/data/data.php.

*Author contribution*. X.-Y. Yang designed this study and analyzed data. All authors contributed to the interpretation of the results. X.-Y. Yang wrote the manuscript with input from all authors.

*Competing interests*. The authors declare that they have no conflict of interests.

**Acknowledgments**

We thank ECMWF for providing the reanalysis datasets of ERAinterim and ORA-s4, and NSIDC for providing the sea ice concentration dataset. We greatly appreciated John E. Walsh and the other anonymous reviewer for their insightful suggestions and generous help in improving the quality of this study.

X.-Y. Yang is supported by the Natural Science Foundation of China (Grant 41576178). G. Wang is supported by the Program of Shanghai Academic/Technology Research Leader (17XD1400600) and the National Natural Science Foundation of China (41976003). N. Keenlyside acknowledges support from the NordForsk ARCPATH (award 76654) and European Union's H2020 Blue-Action (grant agreement no. 727852) projects.

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

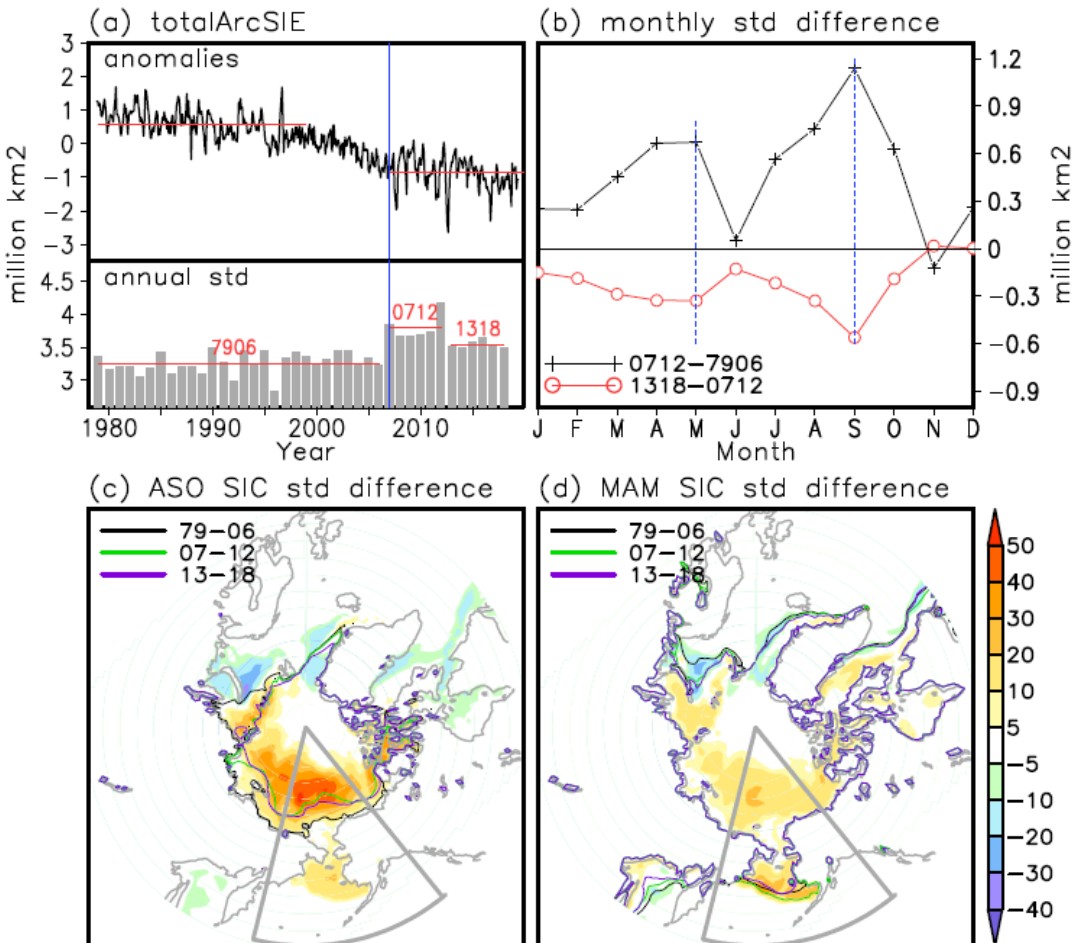

**Figure 1.** (a) monthly total Arctic sea ice extent (totalArcSIE) anomalies (upper panel) and annual standard deviation (lower panel) during the period of 1978-2019. Red lines denote the mean values for different periods. Vertical blue line indicates the year of 2007. (b) differences of multi-year averaged standard deviations (std) for the different calendar months for the three periods designated in (a) lower panel. Black line denotes the std values of 2007-2012 mean minus 1979-2006 mean. Red line denotes the std values of 2013-2018 mean minus 2007-2012 mean. Blue dashed lines are the two peak months, September (thick) and May (thin). (c) Sum of absolute differences in the August-October mean sea ice concentration (SIC) multi-year averaged standard deviation between the three periods (i.e., the black line peak values minus the red line peak values in (b)). Solid contours marks the climatological sea ice edges (SIC=15%) for three periods. The Pacific sector (Chukchi-Bering) is delimited by the gray fan-shaped frame. (d) Same as (c), but for March-May mean.

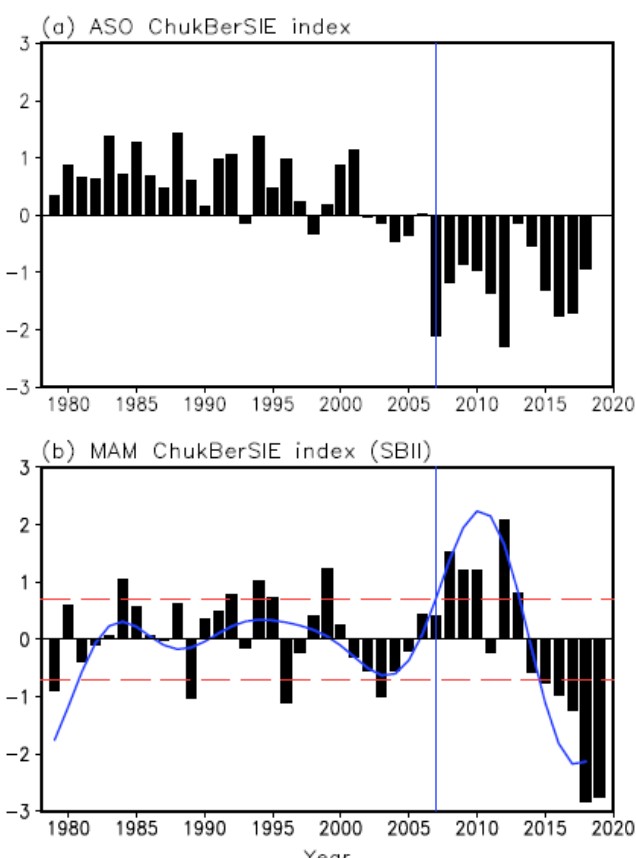

**Figure 2.** (a) Normalized summer (August-October mean) ChukBerSIE index. (b) Normalized spring (March-May mean) ChukBerSIE index (SBII, black bars), superposed by its decadal component (blue lines). Red dashed lines mark the threshold of ±0.7 standard deviation for selecting the composite years. Vertical blue line indicates the critical year of 2007.

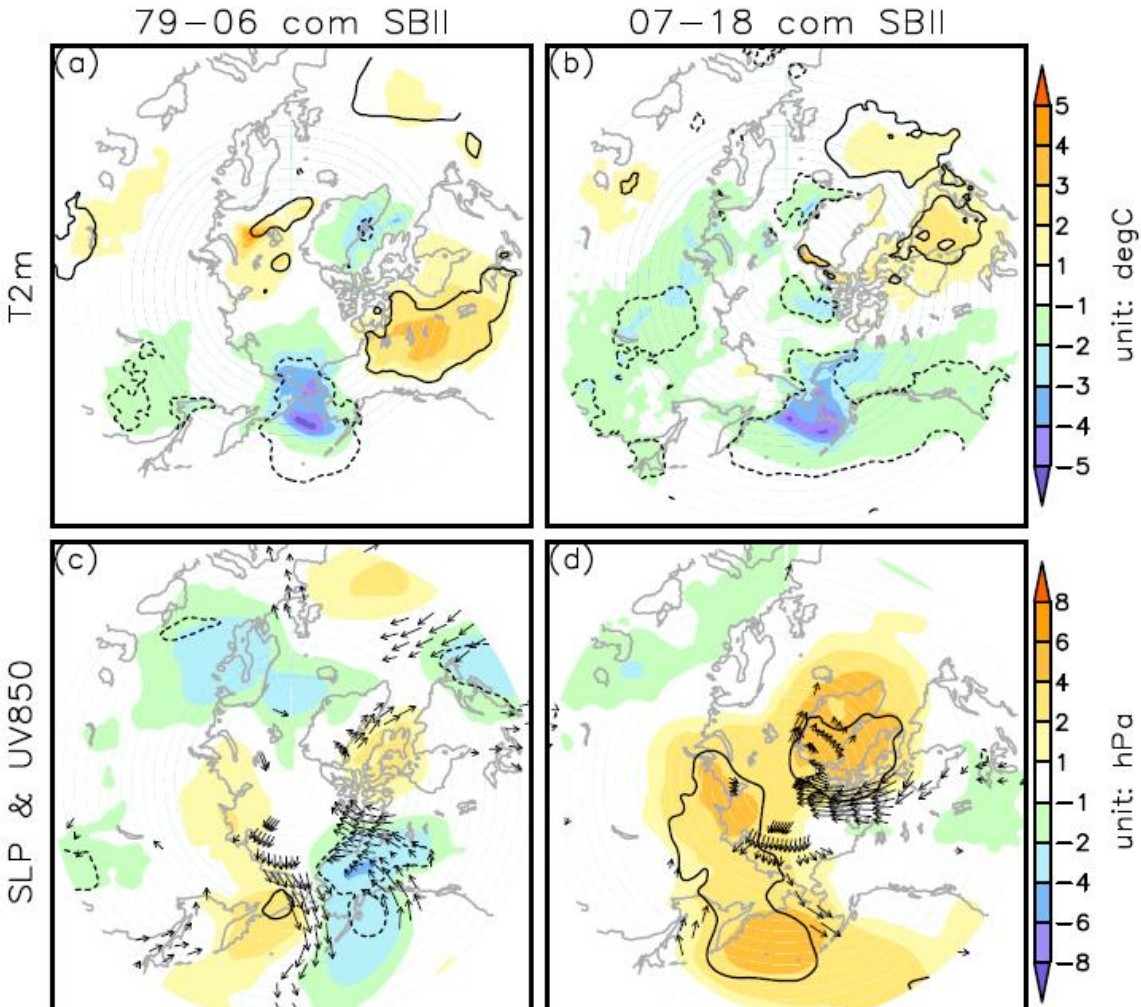

**Figure 3.** Composites of March-April-May (a)&(b) 2-metre air temperature anomalies; (c)&(d) sea level pressure (color shading) and 850hPa wind (vectors) anomalies based on the SBII high minus low index during the prior-2007 period (left panels) and the post-2007 period (right panels). Solid (dashed) lines enclose the positive (negative) values that are significant at the 95% confidence level. For 850hPa wind vectors, only the meridional wind anomalies that significant at 95% confidence level are presented.

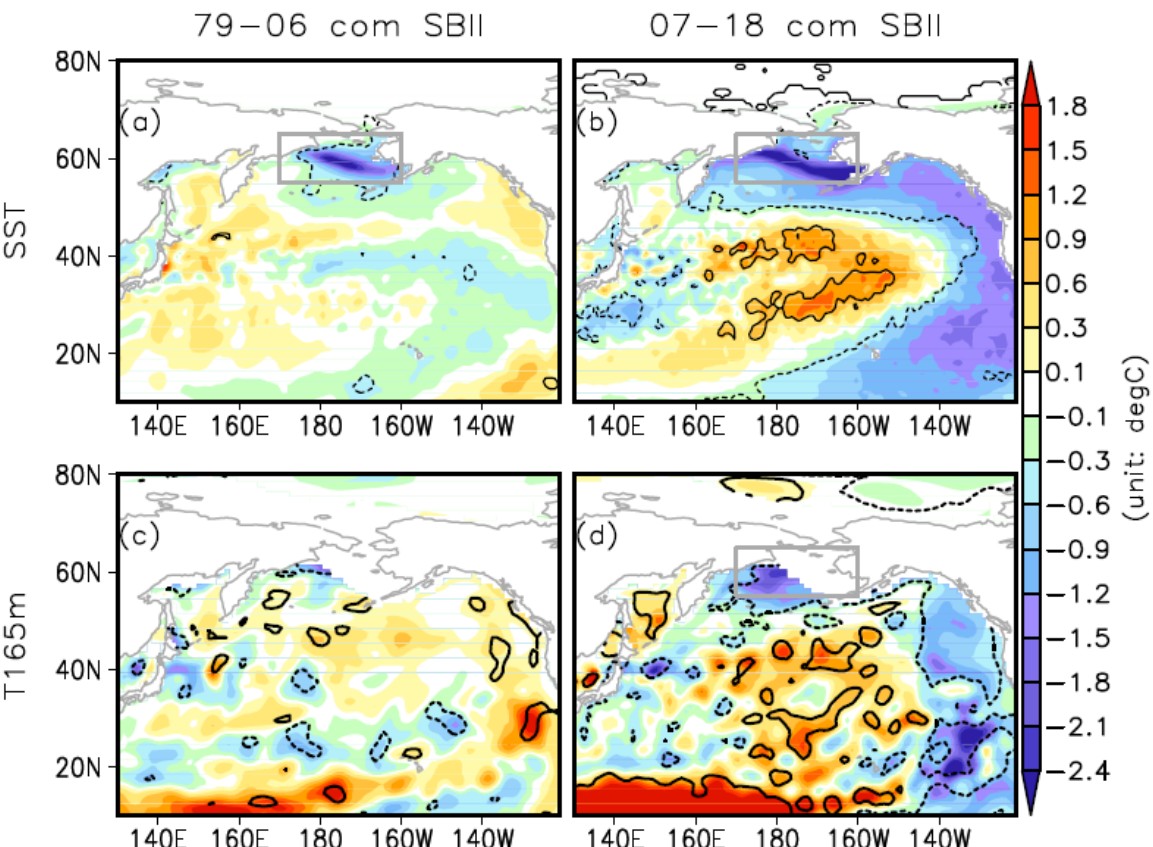

**Figure 4.** Composites of March-April-May (a)&(b) sea surface temperature anomalies; (c)&(d) subsurface (165m) water temperature anomalies based on the SBII high minus low index during the prior-2007 period (left panels) and the post-2007 period (right panels). Gray rectangle defines the area of Bering Sea (170-200E, 55-65N). Solid (dashed) lines enclose the positive (negative) values that are significant at the 95% confidence level.

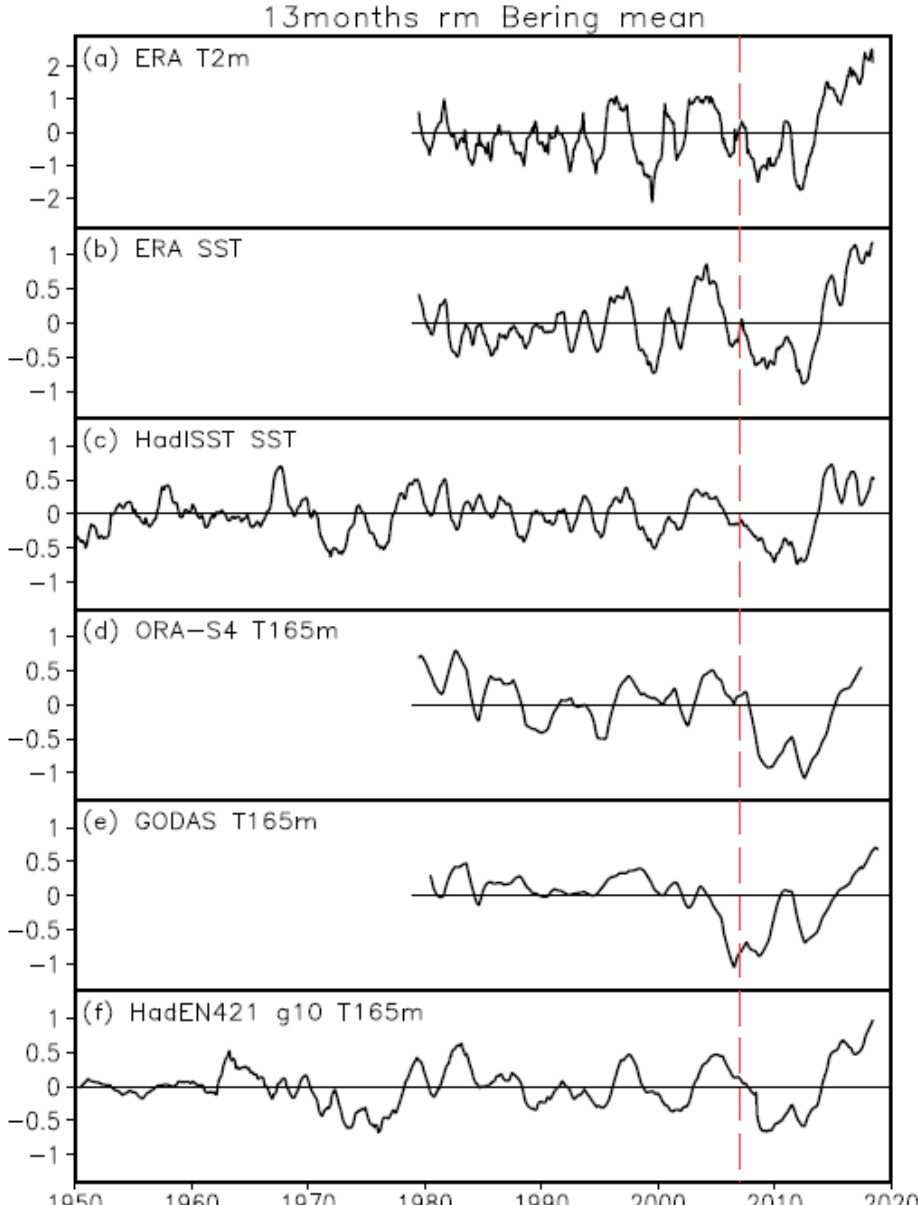

**Figure 5.** Time series of Bering-mean temperature anomalies from various datasets. (a) ERAinterim 2-metre air temperature; (b) ERAinterim sea surface temperature; (c) HadISST1 sea surface temperature (with the global mean SST subtracted from the original data); (d), (e) & (f) subsurface (165m) temperature from ORA-S4, GODAS and HadEN4.2.1 g10 datasets. All data is low-pass filtered by 13 months running mean. Dashed red line marks the critical year of 2007.

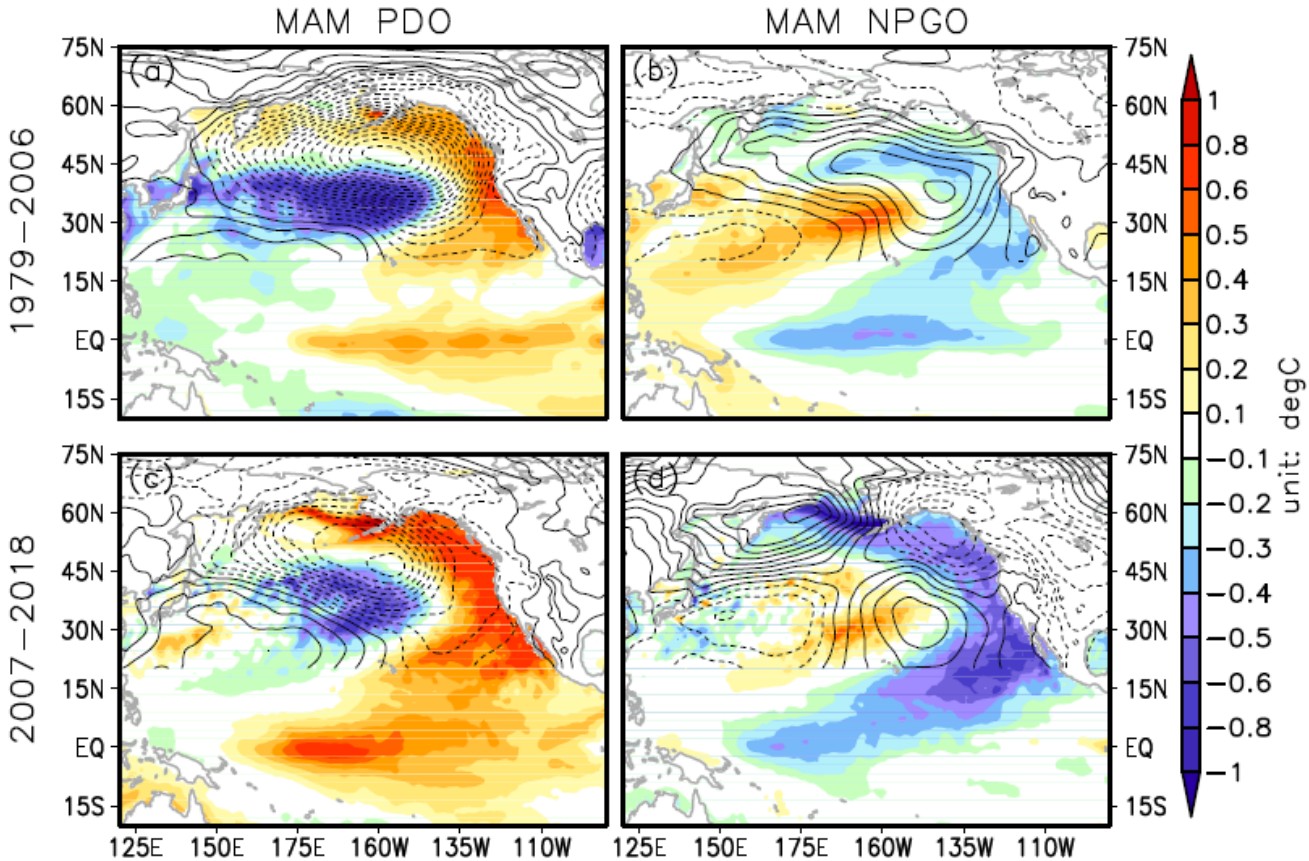

**Figure 6.** (a) Regression of March-May mean sea surface temperature anomalies (color shading) and sea level pressure anomalies (contour) on the PDO index during the period of 1979-2006. The solid and dashed contours indicate the positive and negative anomalies of SLP. The contour interval is 0.2hPa. (b) Same as (a), but regression on the NPGO index. (c) & (d), same as (a) & (b), but during the period of 2007-2018.

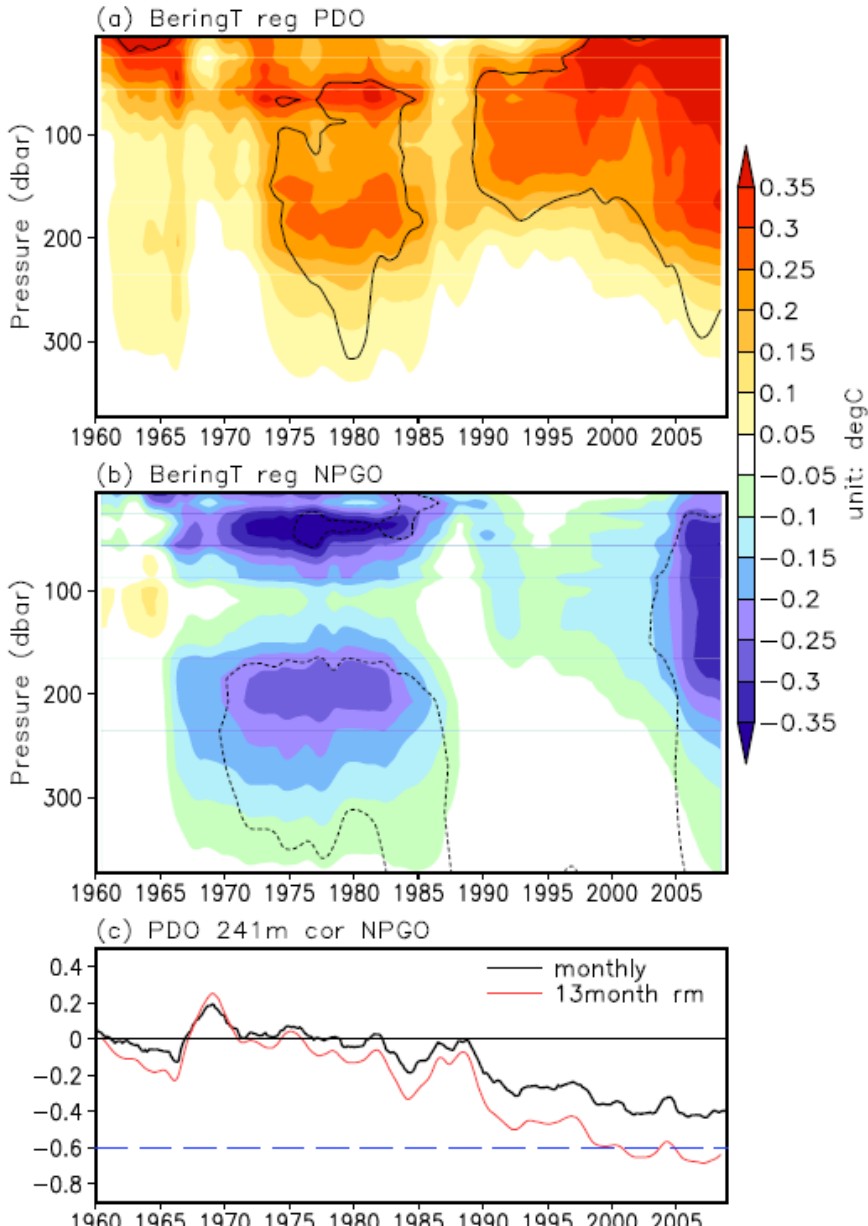

**Figure 7.** 241-month running regression of the HadEN4 subsurface Bering sea temperature anomalies on the PDO index (a) and the NPGO index (b). All the data are preprocessed by 13 months low-pass filter. Solid (dashed) lines enclose the positive (negative) regression coefficients that are significant at the 95% confidence level. (c) 241-month running correlation between the PDO and NPGO indices. Black (red) lines indicate the original monthly (13 months low-pass filtered) correlation.

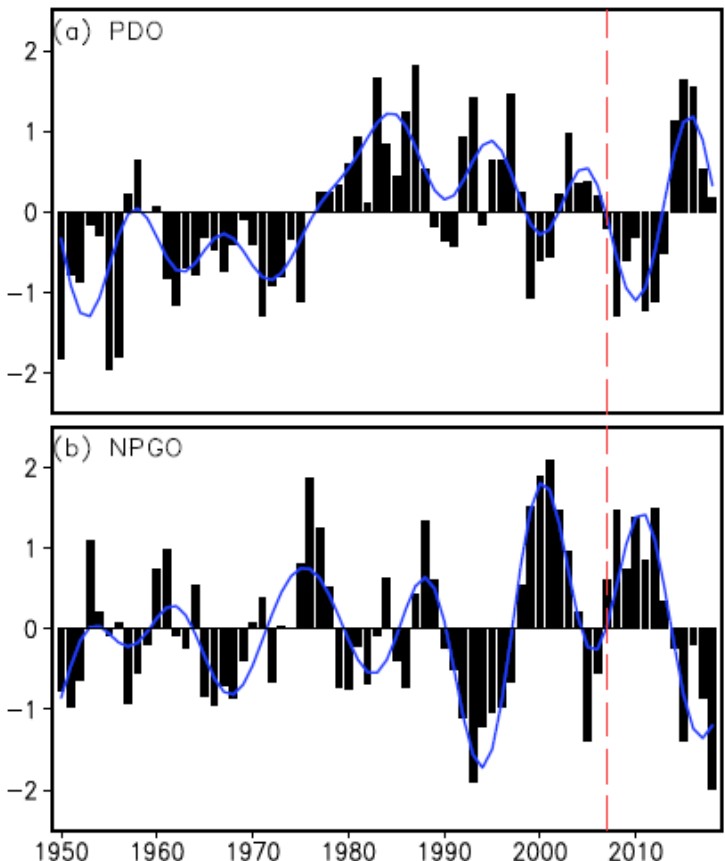

**Figure 8.** (a) Time series of the annual mean PDO index (black bar) superposed by its decadal component (blue line) during the period of 1950-2018. The red dashed line indicates the critical year of 2007. (b) Same as (a), but the NPGO index.

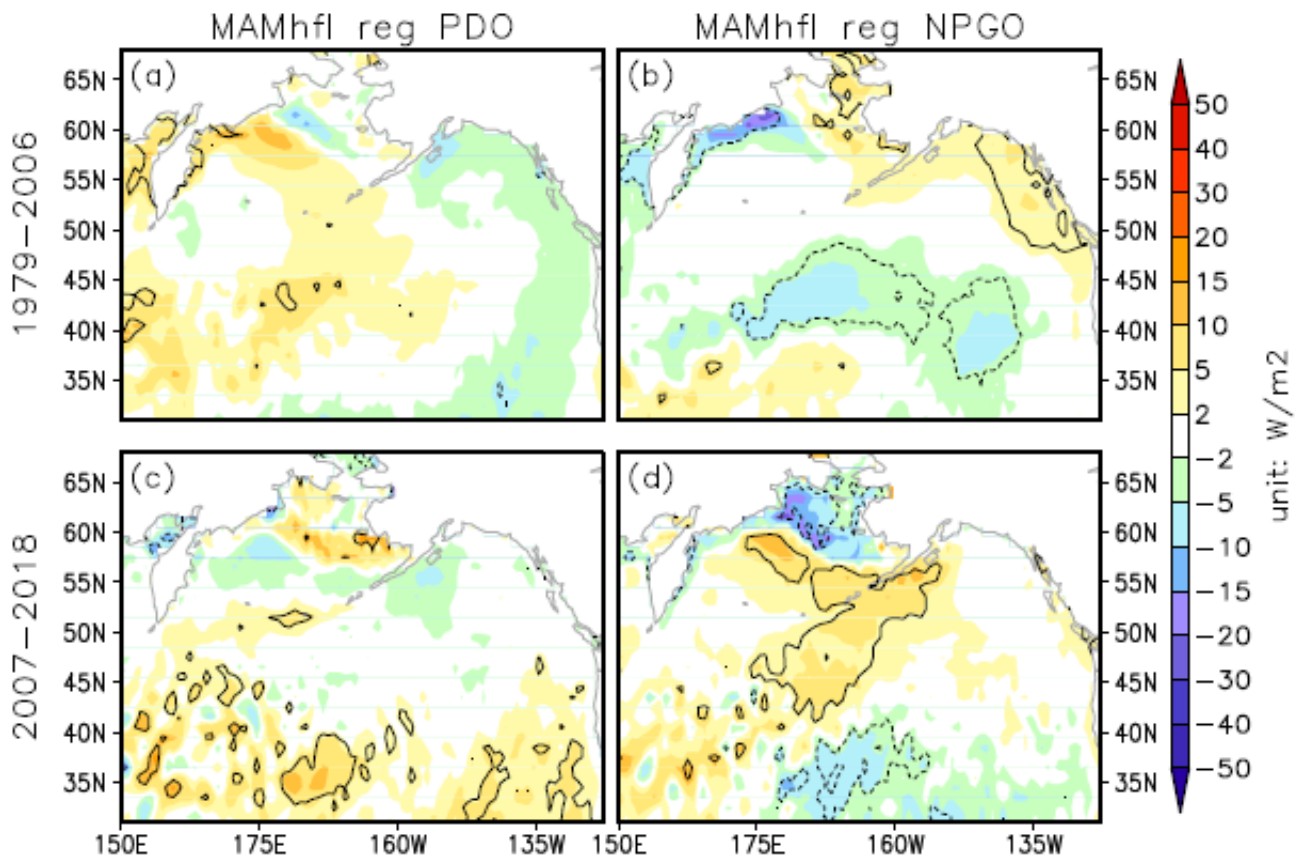

**Figure 9.** (a) & (b) Regression of March-May mean surface turbulent (sensible + latent) heat flux onto the PDO (left panels) and NPGO (right panels) indices during the period of 1979-2006. (c) & (d) Same as (a) & (b), but for the period of 2007-2018. Solid (dashed) lines enclose the positive (negative) regression coefficients that are significant at the 95% confidence level.

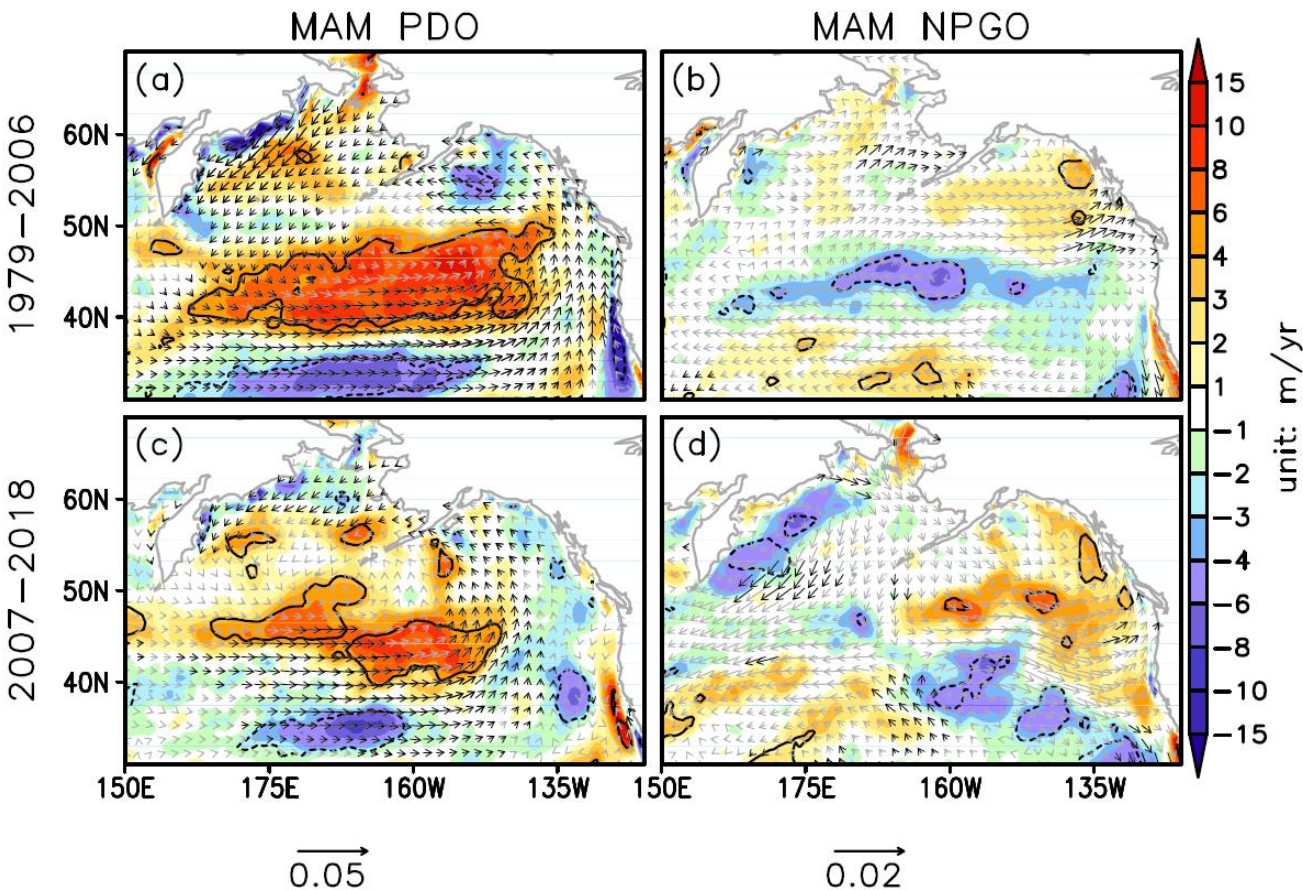

**Figure 10.** Same as Fig. 9, but for the regression of surface wind stress (vector) and Ekman pumping rate (color shading) anomalies. Solid (dashed) lines enclose the positive (negative) regression coefficients of Ekman pumping that are significant at the 95% confidence level. The black (gray) vectors indicate the wind stress regression coefficients are statistically significant (non-significant) at the 95% confidence level.

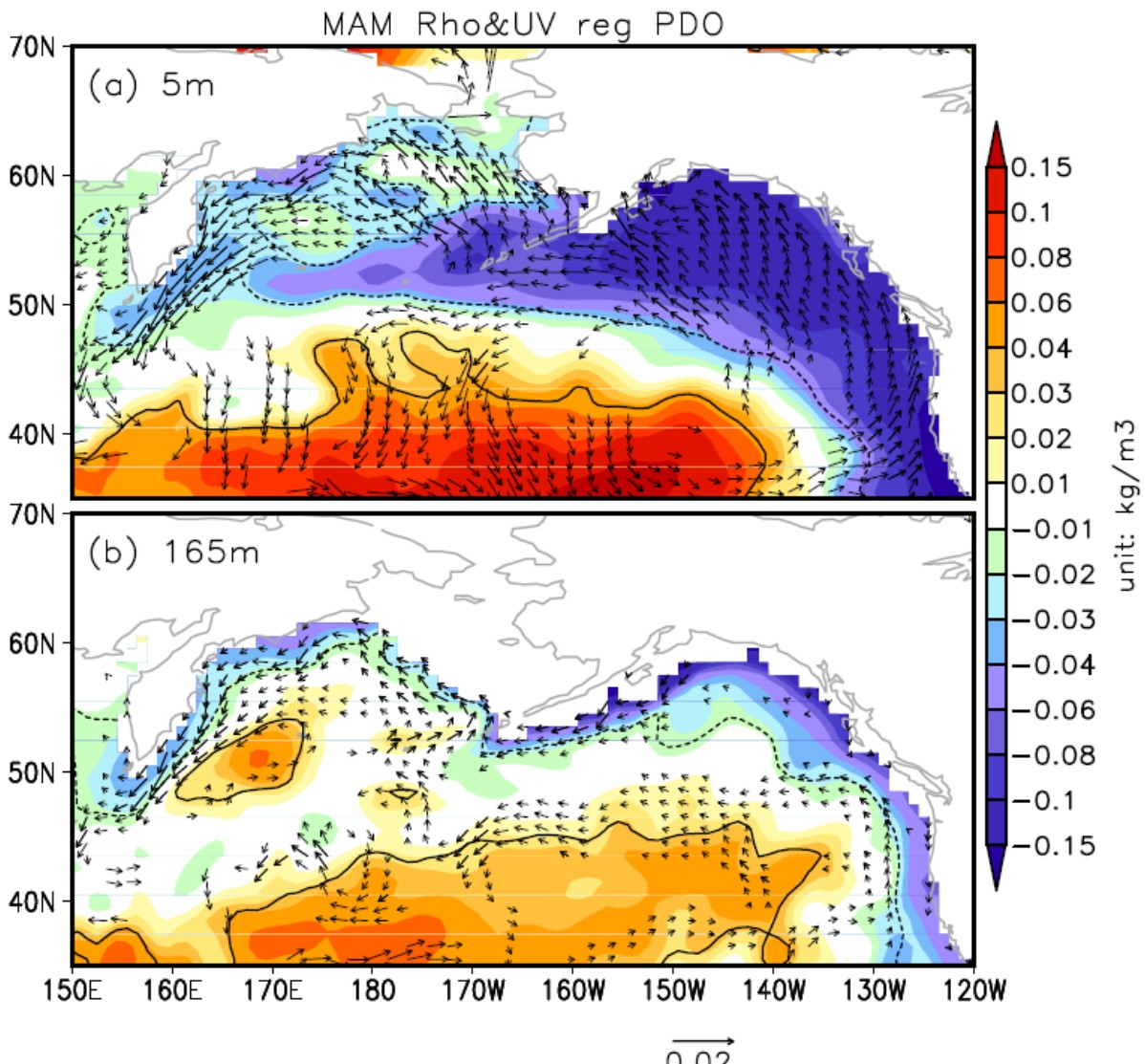

**Figure 11.** (a) Regressions of surface (5m) density (color shading) and zonal and meridional velocities (vector) onto the PDO index. Solid (dashed) lines enclose the positive (negative) density anomalies that are significant at the 95% confidence level. Only the velocity anomalies that are significant at the 95% confidence level are plotted. (b) Same as (a), but for the subsurface layer (165m).

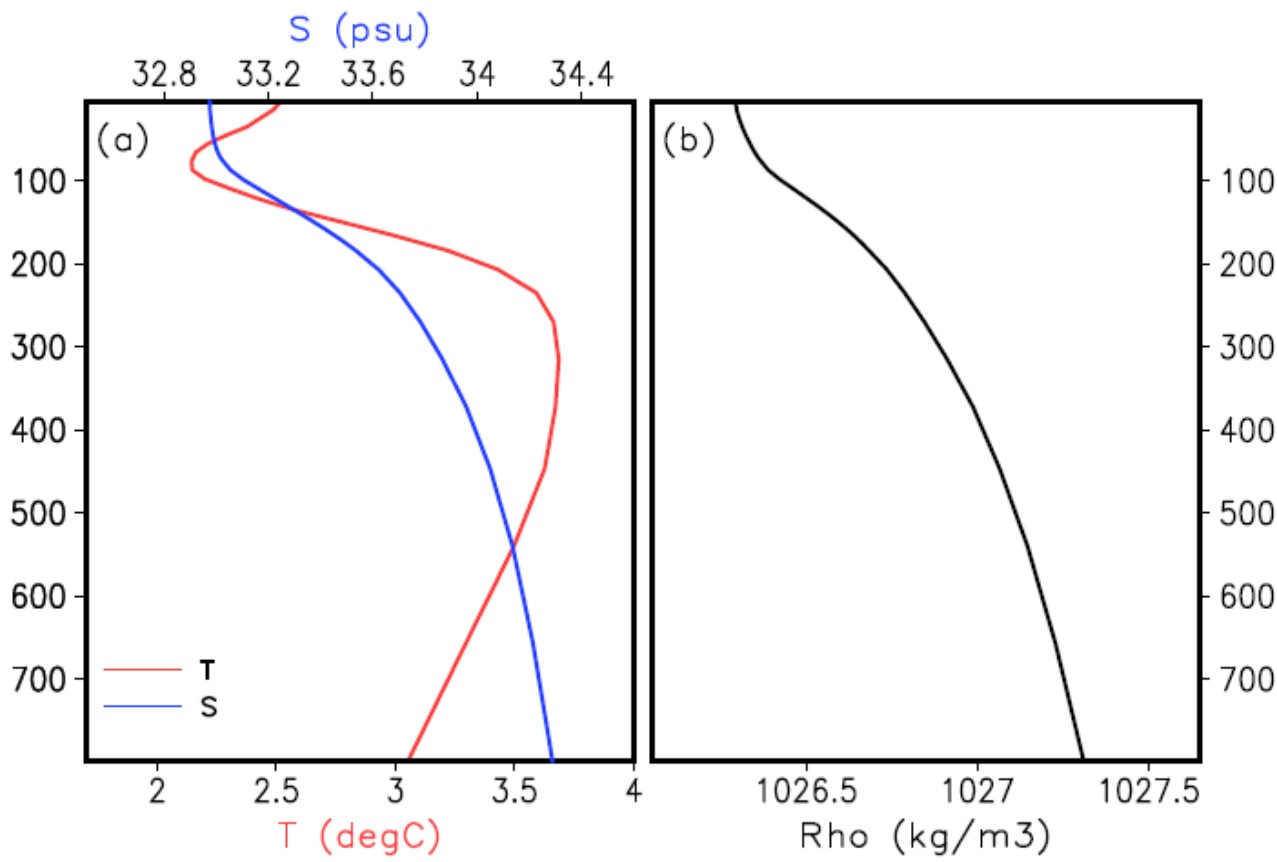

**Figure 12.** Climatological vertical profile of spring (March-May) Bering deep basin (160-190E, 50-60N) mean temperature and salinity (a) and density (b).

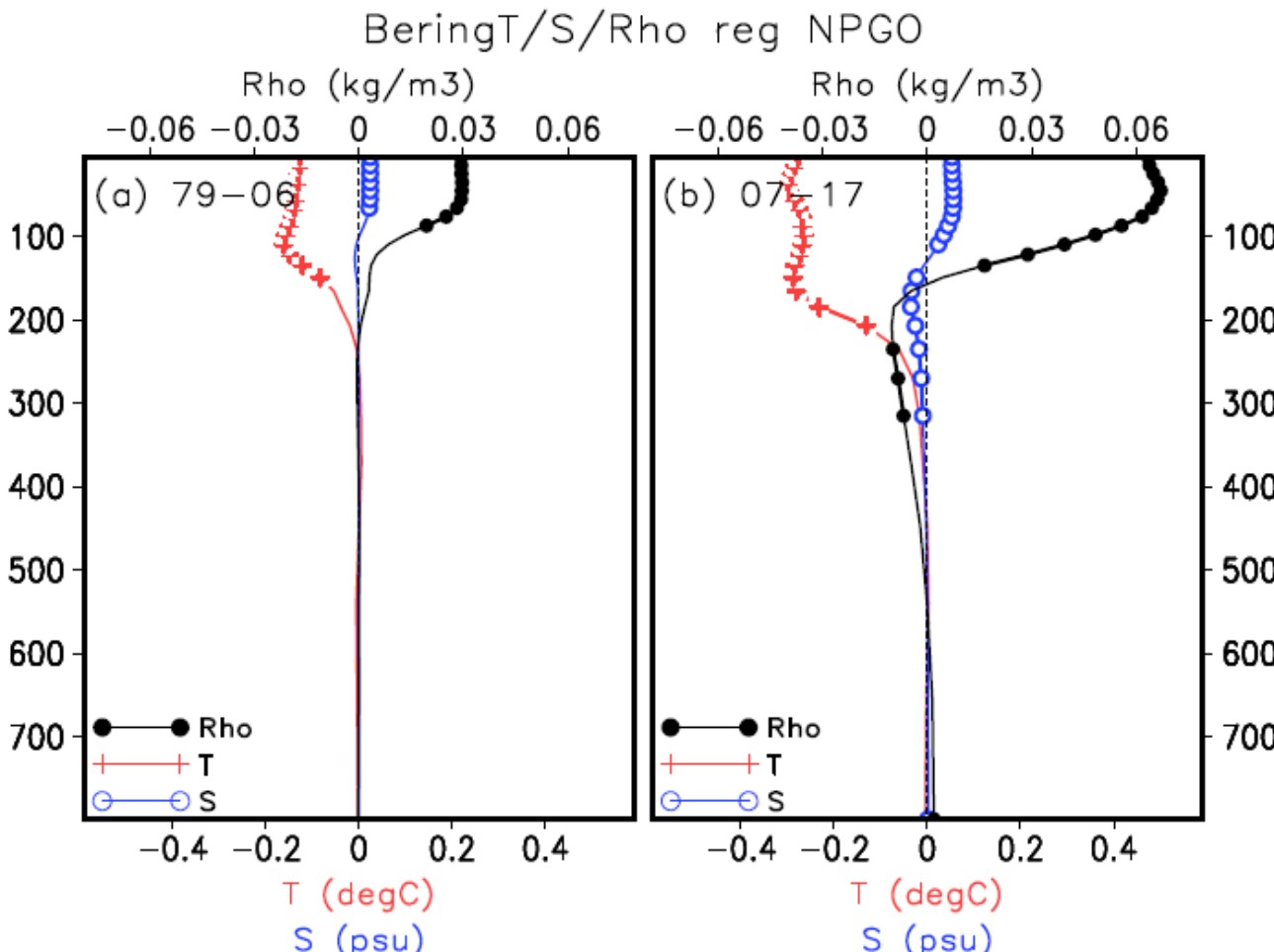

**Figure 13.** Regression of Spring (March-May mean) Bering deep basin temperature, salinity and density on the NPGO index during the 1979-2006 period (a) and the 2007-2017 period (b) for each depth layer. The marks of black dot, red cross and blue circle denote the regression coefficients that are significant at the 85% significant t test for density, temperature and salinity respectively.

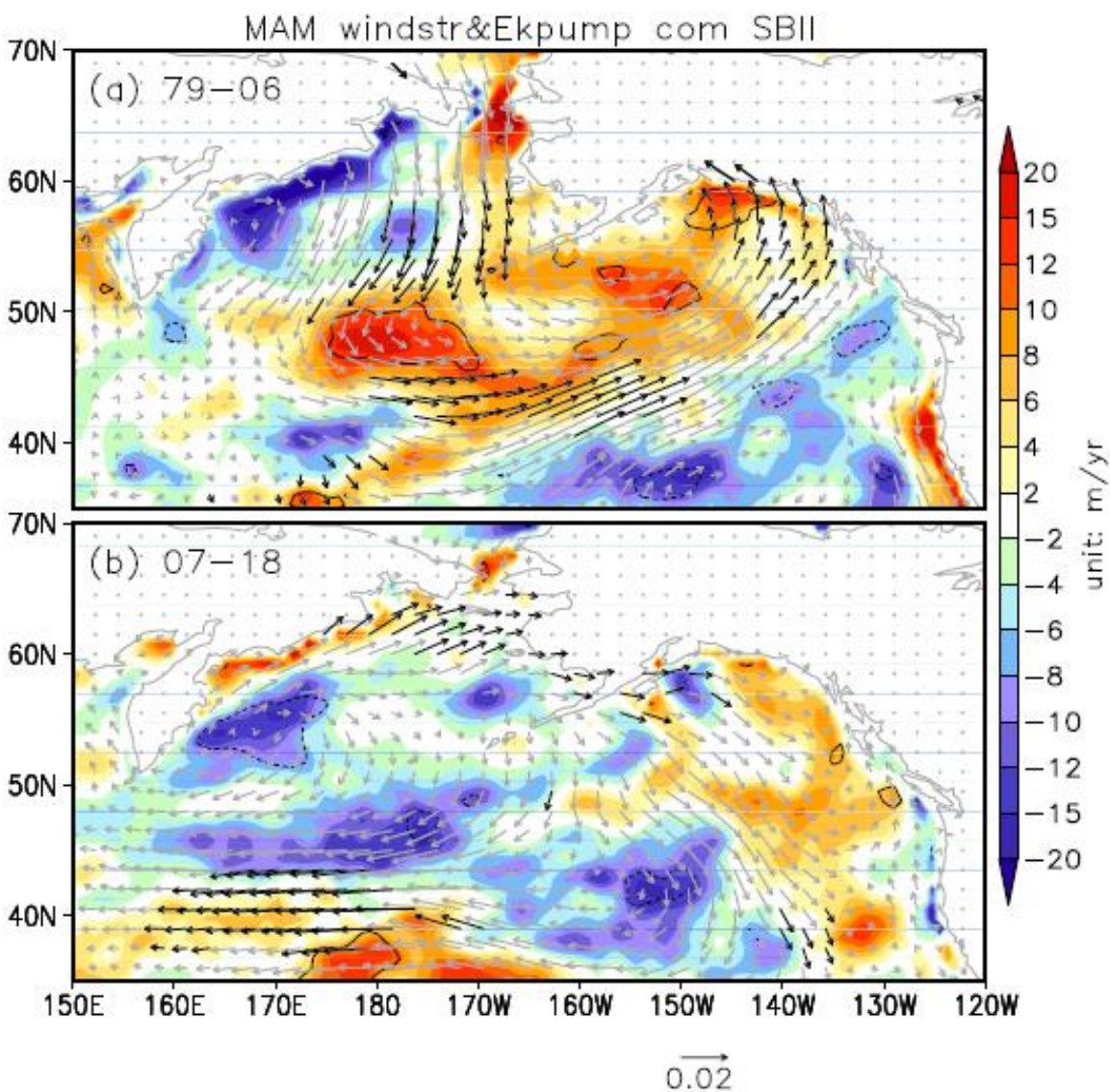

**Figure 14.** Composites of spring wind stress (vector) and Ekman pumping rate (color shading) based on the SBII high minus low index during the period of 1979-2006 (a) and the period of 2007-2018 (b).

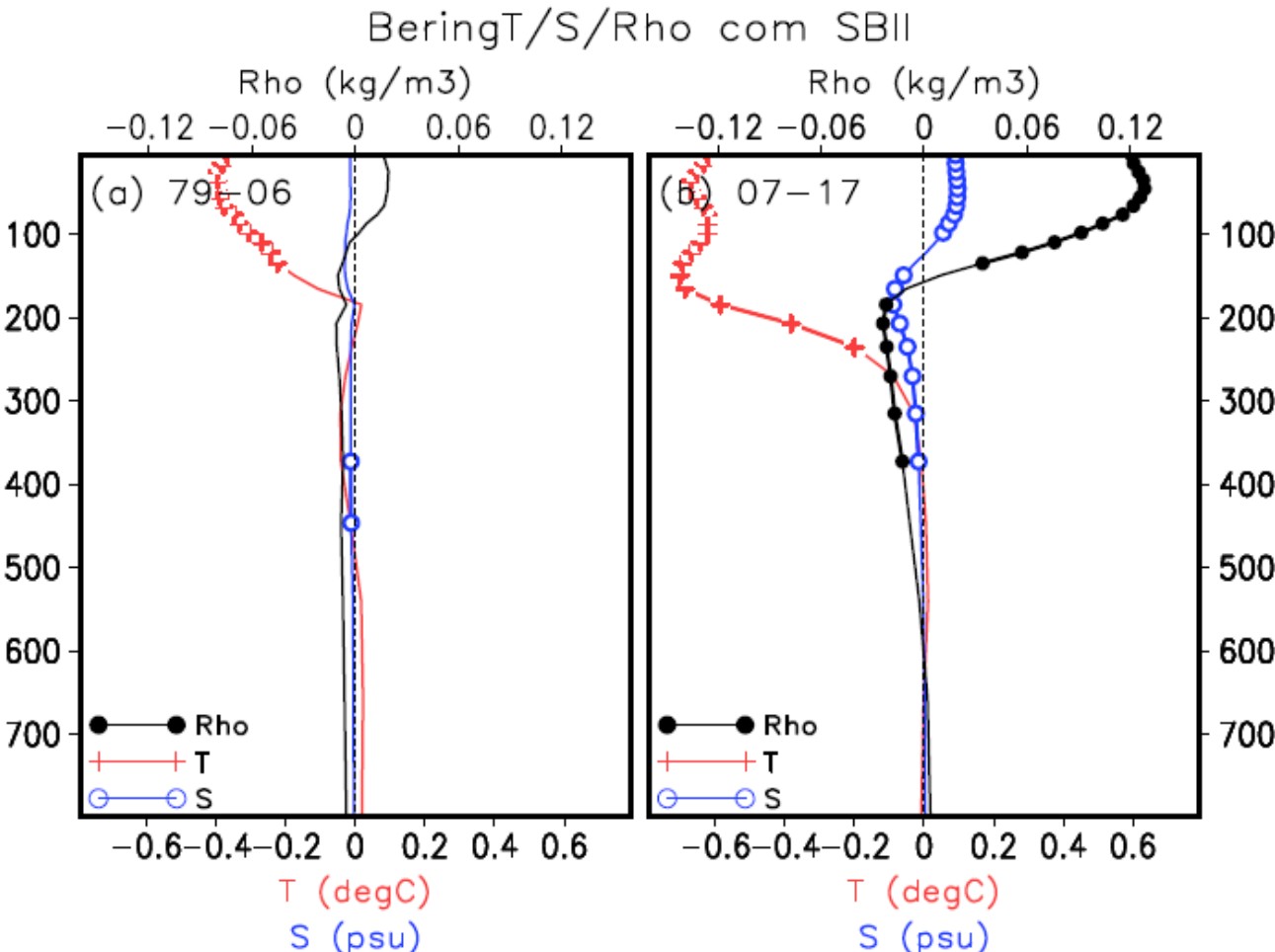

**Figure 15.** Composites of Spring (March-May mean) Bering deep basin temperature, salinity and density based on the SBII high minus low index during the 1979-2006 period (a) and the 2007-2017 period (b) for each depth layer. The marks of black dot, red cross and blue circle denote the regression coefficients that are significant at the 85% significant t test for density, temperature and salinity respectively.