# Peer review of "The Arctic sea ice extent change connected to Pacific decadal variability"

_The Cryosphere, 2019_

## Referee Comment (RC1) · John E. Walsh (Referee) · 9 Jul 2019

This paper shows that the Bering Sea was a major contributor to the increased amplitude of the seasonal cycle of pan-Arctic ice extent in the first decade of the 21st Century. Perhaps more importantly, it addresses a topic that is in need of further understanding: the variability of Bering Sea ice extent over decadal timescales. While the paper contains some interesting results, it comes up short in two respects: (1) recent events have passed it by, and (2) the explanation of the Bering's decadal variations is largely speculative.

With regard to (1), the paper's analysis ends in 2015. Since 2015, the Bering Sea

has been the region of perhaps the most remarkable extremes of sea ice extent in the historical record: the extreme minima of 2018 and 2019. Bering ice extents during these two winters were only about 50% of the previous minima of the satellite era (http://nsidc.org/arcticseaicenews/2019/03/; also see figure below). This represents a spectacular reversal from the positive anomalies of 2008-2013 that shape the results and conclusions of the present paper. The recent extreme minima have put the Bering at the forefront of climate discussions. There are obvious implications for the Bering sea ice trends highlighted in Figure 1 and elsewhere in the paper. As a result, the paper is seriously dated before it even gets into the publication process.

The explanation for the Bering Sea ice history, especially its decadal-scale variability, leaves the reader wondering about the value added. The authors show that there has been a cooling of the Bering sea at depth, coincident with the 2008-1013 positive sea ice anomalies, and that the water temperatures generally correlate well with the Bering ice extent over the entire record. But the reasons for the water temperature variations are only discussed speculatively: there is mention of Bering Strait throughflow (p. 6, lines 12-21) and atmospheric circulation variations (p. 5, bottom – p. 6, top), but there is no rigorous analysis. The text does not even indicate whether there was an increase in the heat exported through Bering Strait in the 2007-2008 timeframe, nor whether there would be a compensating inflow of heat from the south. The hypothesis about changes in ocean currents and Ekman pumping is also not supported by hard evidence, as no ocean current data were included in the analysis. The linkage between ice extent and water temperature does not distinguish cause and effect, and even the possible linkage with the PDO is rather murky in view of the lead-lag correlations in Figure 3d. In short, the paper does not provide a compelling explanation for the Bering Sea winter/spring ice expansion of 2008-2013 (let alone the spectacular abrupt decrease in 2018 and 2019).

Some specific comments:

1. Page 1, line 19: The decrease of pan-Arctic ice extent in recent decades has not

really been "continuous", as there have been ups and downs associated with interannual variability. One can also question the statement in line 20 that "the rate of ice loss accelerates from the 1990s". If one calculates the trend of pan-Arctic September ice extent for the 2007-2018 period, the trend is essentially zero.

2. Page 1, line 32: Not a sentence.

3. Page 4, lines 32-33: In another example of the dated nature of the study, the PDO has been positive for nearly the entire period since 2015.

4. Page 6, lines 1-4: Figure 7 needs some elaboration, including an explanation of how Ekman pumping contributes to the proposed linkage between the atmosphere and the Bering Sea water temperatures and sea ice. In particular, what happens dynamically when Ekman pumping changes from negative to positive (color bar in Figure 7)? What causes the sign reversal, and how do upwelling and downwelling relate to changes in water temperature? In other words, please present more information on the mechanistic linkage alluded to here.

5. Page 6, lines 22-31: There should be some description of the NPO and NPO-NPGO, and some supporting references. As it now stands, the NPO and NPGO seem to appear out of the blue.
* * *
[Figure]

**Fig. 1.** Bering Sea ice extent through 2019

---

## Referee Comment (RC2) · Anonymous Referee #2 · 18 Jul 2019

Review of "The recent amplifying seasonal cycle of the Arctic sea ice extent related to the subsurface cooling in the Bering Sea" by Yang and Wang

Summary

This paper presents an analysis of Bering and Chukchi Sea ice, showing contrasting decadal variability with decreases in the summer-autumn and increases in the winter-spring. Sea ice, atmospheric reanalyses, and ocean temperatures and velocities are examined to explain this behavior. The authors find that the pattern is likely due to subsurface cooling in the northern Bering Sea.

General Comment

[Figure]

Overall, this is well-written paper and the analysis is competent. However, I have a few notable issues, listed below. The second issue in particular is critical and is the major reason for the decision of Major Revision.

1. The authors generally treat the Bering and the Chukchi Seas (B-C) as one system, but I'm not sure if this really appropriate. For example, they say that the B-C shows decreases in the summer-autumn and increases in the winter-spring. However, the Bering is ice-free during summer-autumn, so the decrease is in the Chukchi, not really the B-C. Similarly, the Chukchi is completely ice-covered during winter-spring, so the increase there is really in the Bering, not the B-C.

During the analysis, they do start to treat the two more separately, but their approach is generally that these are two closely inter-connected systems and I'm not convinced that this is the case. There is indeed connection through the Bering Strait, and though that is narrow and shallow, there has been much research demonstrating that inflow of warm Pacific waters have contributed to summer ice loss in the Chukchi. But the Chukchi is also connected to the rest of the Arctic Ocean, particularly the neighboring seas – the Beaufort and the East Siberian. For example, the Beaufort Gyre circulation transports ice into the Chukchi and much of that ice is thicker, multi-year ice. This aspect seems to be ignored in the paper – particularly the role of changes in multi-year ice, which has been rather dramatic in recent years.

And while the Bering influences the Chukchi, I don't think the reverse is generally true – the ocean waters flow into the Chukchi. There can be some sea ice outflow, but this is pretty rare and ephemeral. So, when talking about the B-C, it's really more of a one-way street, not an interacting system. The main connection would be through the atmosphere where circulation patterns can affect the transport of heat, etc. between the two regions.

Maybe this is more semantics on my part. In the body of the paper, the authors seem to be clearer and focus more on the Bering Sea and then interaction with the Arctic

Ocean in general. So maybe some changes in phrasing in places would help here.

2. The other main issue is, unfortunately, the timing of this paper. As the authors are probably aware, the situation in the Bering has completely reversed over the last two winters. Both 2017-2018 and 2018-2019 were record or near-record lows over most of the winter-spring. I recognize that things can change and that papers have to be published at some point and new data will come in after submission. However, this paper stops with 2015, so after the last two winter-springs, it feels particularly out date. I know data isn't necessarily immediately available, but for example sea ice data is provided in near-real-time, as is ECMWF. I'm not sure about the ORAS4 ocean temperature and velocity data, but it appears more recent data is available: https://www.ecmwf.int/en/research/climate-reanalysis/ocean-reanalysis. Even without updated ocean data, at least some of the analysis could be updated.

Because the change is so substantial, I feel the authors should really redo the analysis through at least 2018. As noted above, I recognize that at some point, analysis has to stop and a manuscript must be prepared and submitted. But this paper just feels too out of date to be particularly relevant. At the very least, if the authors can't update their analysis, they should thoroughly discuss the recent changes in the Bering Sea ice cover and the implications for their study.

3. Overall, the analysis is rather cursory. The authors pull together the right data sets and the analysis that they do is reasonable. But it doesn't seem enough to really substantiate their hypotheses - particularly in light of #2 above. The paper is <6 pages long and while I enjoy short, concise papers, the analyses are not very deep and they don't dig into the data enough to definitively support their conclusions.

4. The final issue is several grammar errors – not major, but occur occasionally throughout the manuscript. Some are note below in Minor Comments, but a thorough proofreading for English grammar would be helpful.

Specific Comments (by page and line number)

5, 16-17: "significant" and "insignificant" are each used. Do these refer to statistical significance? If so, the significance level should be specified. If not, then I would suggest different wording, e.g., substantial, notable, etc. to avoid confusion over the use of "significant".

Figure 1b: I find this figure a bit confusing. The 120W-60W is not labeled – that's the Canadian Archipelago. I recognize this isn't terribly relevant to the study, but odd to have just that longitude range unlabeled. More relevant though is: where is the Laptev? I think it's included in what is labeled "East Siberian" – if so, that should be noted. Maybe what would help is a map that has the seas labeled and the longitude boundaries outlined.

Minor Comments (by page and line number)

1, 9: "in recent decade" – a minor grammar issue, but it makes things ambiguous: is it "in the recent decade" (i.e., the last 10 years), or is it "in recent decades" (i.e., 20+ years)? This is repeated elsewhere in the manuscript as well.

1, 27: "transit" should be "transition"

2, 10: "urgent" not "urgently"

2, 11: remove "the" before "Arctic climate change"

5, 22: "decline" not "descent"

6, 15: as noted above, I don't think you can any longer say "lengthening sea ice seasons" for the Bering – certainly not for the last two years. 6, 29: In light of the last two years' worth of data, I think the question of whether the "amplifying seasonal cycle of Arctic sea ice cover will [be] sustained" has been answered, at least in terms of the Bering Sea: no!

---

## Author Comment (AC1) · 25 Aug 2019

This paper shows that the Bering Sea was a major contributor to the increased amplitude of the seasonal cycle of pan-Arctic ice extent in the first decade of the 21st century. Perhaps more importantly, ...... The paper is seriously dated before it even gets into the publication process.

Reply: Thank you for reminding us of this important information! We follow your advice and update all of the available datasets to the latest: (1) NSIDC monthly sea ice concentration data from November 1978 to June 2019; (2) ECMWF ERA-interim meteorological fields data from January 1979 to April 2019; (3) ECMWF ORA-S4 ocean

fields data from January 1958 to December 2017; (4) JISAO PDO index from January 1900 to September 2018; (5) NCEP GODAS ocean assimilation data from January 1980 to May 2019; (6) HadEN4.2.1 g10 ocean subsurface analyses data from January 1950 to December 2018. To probe into the physical mechanism related to the sea ice change, we download two other datasets, i.e., WHOI OAflux data from January 1958 to December 2018 and NPGO index constructed by Di Lorenzo from January 1950 to December 2018. All above changes are described in the revised paper Section 2.

Once total Arctic sea ice extent time series is extended to the latest, its variation magnitude seems drop down again. That is, 2013-2018 annual standard deviation is overall lower than that of 2007-2012, representing a flavor of decadal scale oscillation feature (Fig. 1a). Fig. 1b shows the standard deviation differences for each month. It is obvious that discrepancies between the three periods (2007-2012 minus 1979-2006, and 2013-2018 minus 2007-2012) are primarily come from the summer (Aug-Oct, peak in Sep) and spring (Mar-May, peak in April) seasons. The spatial patterns further show that the Pacific section including the Chukchi sea and the Bering Sea (gray fan) as well as the Laptev and eastern Siberian seas is the major contributor to these decadal changes of total Arctic sea ice extent. The normalized Chukchi-Bering SIE indices in ASO and MAM (fig. 2) both exhibit abrupt change since 2007. In summer this change can be viewed as a mean state mutation or regime shift (fig. 2a), while in spring it is more like a decadal oscillation (fig. 2b). In another words, the substantial retreat of Chukchi sea ice in summer and growth of Bering sea ice in spring occurs synchronously in 2007, collaborating on the soaring up of the totalArcSIE std. The dropping down of annual std since 2013, however, should be primarily attributed to the spring Bering sea ice decline.

We made these revisions in the section 3.1 of the revised manuscript.

Fig. 1 (a) time series of monthly total Arctic sea ice extent (totalArcSIE) anomalies (climatology calculated during 1979-1998, upper panel) and annual standard deviation (lower panel) during the period of 1978-2019. Red lines denote the mean values for

different periods. Vertical blue line indicates the critical year of 2007. (b) monthly differences of standard deviation for three periods designated in (a) lower panel. Black line denotes the values of 2007-2012 minus 1979-2006. Red line denotes the values of 2013-2018 minus 2007-2012. Blue dashed lines are the two peak months, September (thick) and April (thin). (c) Sum of August-October mean sea ice concentration (SIC) standard deviation differences between the three periods (absolute values of 0712-7906 (black lines in fig. 1b) plus absolute values of 1318-0712 (red lines in fig.1 b)). Solid contours mark the climatological sea ice edges (SIC=15%) for three periods. The Chukchi-Bering sector is delimited by the gray fan-shaped frame. (d) Same as (c), but for March-May mean.

Fig. 2 (a) Normalized summer (August-October mean) ChukBerSIE index. (b) Normalized spring (March-May mean) ChukBerSIE index (SBII, black bars), superposed by its decadal component (blue lines). Red dashed lines mark the threshold of $\pm 0.7$ standard deviation for selecting the composite years. Vertical blue line indicates the critical year of 2007.

(2) The explanation for the Bering Sea ice history, especially its decadal-scale variability, leaves the reader wondering about the value added....... In short, the paper does not provide a compelling explanation for the Bering Sea winter/spring ice expansion of 2008-2013 (let alone the spectacular abrupt decrease in 2018 and 2019).

Reply: Thank you for your suggestions! As we noted in the original manuscript, the spring Bering sea ice high index is associated a SLP dipole pattern prior-2007 but an anomalous high pressure thereafter (figure not shown here). Furthermore, the anomalous Bering SST anomalies exhibits a close connection with North Pacific large-scale SST pattern in the recent decade, in great contrast to the isolated Bering SST variability before 2007 (fig. 3a & b). In addition, the SST anomalies can extend down to subsurface layer after 2007 (fig. 3c & d). This remarkable pattern change, together with the period extension of SBII variation from interannual to decadal scale (fig. 2b), suggests that the spring Bering sea ice variability is getting predominant by the ocean dynamical

processes in Pacific. Numerous previous studies asserted that extra-tropical North Pacific climate variability is dominated by two decadal-scale ocean-atmosphere coupled modes, the Pacific Decadal Oscillation (PDO) (Mantua et al. 1997) and the North Pacific Gyre Oscillation (NPGO) (Di Lorenzo et al. 2008). Though defined independently as the first and second basin-scale EOF modes of sea surface temperature anomalies as well as sea surface height anomalies in the north Pacific Ocean respectively, the PDO and NPGO bear strong resemblance in their physical mechanisms and climate effect. They both are forced by the El Nino-like tropical Pacific SSTA (Newman et al. 2003; Di Lorenzo et al. 2010), feedback to the western Pacific Kuroshio-Oyashio extension through the ocean Rossby wave propagation (Schneider and Corneulle 2005; Ceballos et al. 2009), and impact the low-frequency variability of marine ecosystem system (Miller et al. 2004; Di Lorenzo et al. 2013; Sydeman et al. 2013). Besides the remote tropical ENSO forcing, the PDO and NPGO can be driven by the local ocean-atmosphere interaction. The ocean, as a low-frequency filter, integrates the atmospheric stochastic noise and trigger the decadal-scale oscillation (Newman et al. 2016; Yi et al. 2015). The corresponding large-scale atmospheric modes are the Aleutian Low variability (for PDO) and the North Pacific Oscillation (for NPGO).

In the light of previous study fruits, we infer that the SSTA pattern shown in fig. 3b may relate to the pacific decadal variability. Fig. 4 shows the spring PDO and NPGO patterns for the two periods. The PDO positive phase is associated with the deepening of Aleutian Low, warming in the northeast of Pacific including the Gulf of Alaska and the California coast, paralleling cooling in the Kuroshio extension. Moderate warming in the eastern equatorial Pacific is discernable (Fig 4a). In the recent decade, however, the PDO SLP anomalies abate but the eastern Pacific warming enhances greatly. Notably the Bering Sea and the central equatorial Pacific stand out to be among the most significant warming regions (Fig 4c). The spring positive NPGO corresponds to an anomalous high pressure over the most of North Pacific and negative SLP anomalies over the eastern Asian and Alaska in the early period. The SSTA spatial pattern looks like the PDO SSTA pattern shown in fig 4a but with a quarter phase difference (fig.

4b), which is an allusion of orthogonality between these two modes. The change of NPGO pattern after 2007 is conspicuous: The quadruple SLP anomalies coupling the significant cooling extending all the way from the Bering Sea, the northeastern Pacific to the central Pacific (fig. 4d). It seems that both the PDO and NPGO contribute to the Bering SST anomalies in this decade.

To reaffirm the impact of PDO and NPGO onto the Bering Sea temperature, the running regression is computed from Jan1950 to dec2018 as shown in fig. 5a & b. The Bering temperature indeed exhibits close connection with PDO since the mid-1970s. The correlation first appears in the subsurface layer, then enhances and extends upward with the time evolution, reaching its peak near the surface after 2000. The impact of NPGO, however, is significant only in the 1970s-1980s and after 2005. Fig. 5c shows the 241-month running correlation between the PDO and NPGO indices. There is almost zero correlation before 1990s and significant negative correlation in the new century. By scrutinizing the annual mean PDO and NPDO time series (fig. 6), one can easily perceive increasing synchronism of decadal-scale phase change in recent years. The PDO period is obviously shortened from the traditional 20-40 years to around 10-15 years after 1990s. The NPGO, on the other hand, maintains its periodicity stable, but the oscillation magnitude is almost doubled from the end of 1980s. The strengthening of NPGO and the PDO coalesce to form the apparent anti-resonance after 2007. The synchronization of PDO and NPGO and their common effects on the Bering Sea temperature since 2007 may account for the recent decadal oscillation of spring Bering Sea ice extent.

The robust cause and effect should be based on the establishment of physical linkage. We thereafter explore how the PDO and NPGO modes collaborate to the significant Bering sea temperature anomalies in the recent decade. In spite of high resemblance of SSTA patterns (fig. 4 c & d), great contrast of their SLP patterns between these two modes suggests different atmospheric forcing and consequently distinct oceanic physical processes. Firstly, the surface turbulent heat flux anomalies are regressed on

PDO and NPGO indices (fig. 7). There is hardly significant heat flux anomaly in the Bering area with positive one standard deviation of PDO for the whole period (fig. 7 a & c). The positive phase of NPGO corresponds to the reduced upward heat flux along the northwestern coast of Bering in the early period (fig. 7b). In the later period, the positive and negative anomalies of heat flux appear in the southern and northern Bering Sea respectively, with the zero line approximately along the climatological spring Bering sea ice edge (fig. 7d). Referring to the quadruple SLP pattern (fig. 4d), the anomalous atmospheric cold advection prevails over the Bering Sea, leading to the drop-down of atmospheric surface temperature, then increased heat loss over the open ocean and the negative anomalies of SST. The sea ice expansion follows the decreasing temperature and insulates the ocean-atmosphere exchange, resulting in the negative anomalies of heat flux over the ice cover. This thermodynamic process emerges in the later period, primarily owing to the atmospheric SLP quadruple pattern.

Besides the thermodynamic effect, the NPGO SLP pattern change may play a role in dynamical adjustment of the Bering Sea through its effect on wind stress and Ekman pumping rate. Therefore, the spring wind stress and Ekman pumping rate anomalies are regressed on PDO and NPGO indices respectively (fig. 8). In response to the deepening of AL (fig. 4 left panels), positive phase of PDO is associated with an anomalous cyclonic circulation over the north Pacific without notable decadal change. The relevant positive Ekman pumping anomalies mainly reside in the north Pacific open ocean, while the enhanced northeasterly prevails over the entire Bering Sea (fig. 8a & c). In contrast, the NPGO can hardly cast a spell on the large-scale circulation. Nevertheless, the scattered significant wind stress anomalies in the later period organize an anomalous anti-cyclonic pattern, hence the negative Ekman pumping rate (downwelling) in the western Bering Sea (fig. 8d). The vertical velocity induced by surface wind stress curl usually results in the displacement of thermocline or pycnocline, followed by the dynamical adjustment of the subsurface ocean.

The oceanic density and zonal and meridional velocity anomalies are regressed on the

PDO index (fig. 9). The large-scale cyclonic wind stress anomalies associated with the PDO positive phase (fig. 8 left panels) lead to the strengthening of Alaska stream and enhanced northward transport of heat along the Bering slope, contributing to the surface warming and density decrease in the Bering Sea (fig. 9a). In the subsurface layer, however, the wind-driven component abates rapidly with depth. The anomalous heat transport is limited to the boundary current around the southern Bering deep basin (fig. 9b). Therefore, the Bering Sea warming associated with PDO peaks in the mixed layer (fig. 5a). The circulation anomalies corresponding to NPGO is also investigated, but without any remarkable signal of current and heat transport anomalies (figure not shown).

As the NPGO leads to the anomalous Ekman pumping (fig. 8d) and subsurface cooling (fig. 5b), we examine the pycnocline displacement and vertical exchange between the mixed layer and subsurface ocean in the Bering deep basin (160-190E, 50-60N). The climatological vertical profile of Bering basin (BB) mean temperature (T), salinity (S) and density (Rho) in spring is presented in fig. 10. The BB vertical stratification depends on salinity, with the highly matching of the position of pycnocline and halocline at about 100∼300m depth. In striking contrast to the increase of salinity and density with depth, the temperature profile exhibits a sandwich-like pattern. There is a sharp rising of temperature from ∼2.5°C to ∼3.5°C in 100-300m, then slowly drop down with depth below the permanent pycnocline. It is clear that the cold and fresh water in the mixed layer is superposed over the warm and saline water in the pycnocline. The spring BB T/S/Rho anomalies associated with positive NPGO for different periods are shown in fig. 11. In the early period, NPGO-related changes of water property are confined in the mixed layer. In the later period, the T/S/Rho anomalies associated with positive one standard deviation of NPGO almost doubled in the mixed layer, and the change in the pycnocline can be detected clearly. The colder and saltier water in the surface overlies the less cold and fresher water in the subsurface, corresponding to the density increase (decrease) in the mixed layer (pycnocline).

A NPGO-related atmosphere-ocean-ice feedback mechanism may be inferred as following: The quadruple SLP pattern in the recent decade (fig. 4d) favors the atmospheric cold advection and anticyclonic wind stress curl (fig. 8d) over the Bering Sea in association with the positive phase of NPGO. In response to atmospheric thermal forcing, ocean heat loss enhances in the open water (fig. 7d), leading to the greater surface cooling. On the other hand, the Bering sea ice cover expands with the cooling condition. Ice expansion is usually followed by surface salinification in the adjacent open water owing to brine-rejection effect. Surface water is getting denser with cooling and salinification anomalies. Triggered by the anticyclonic wind stress curl over the BB region, anomalous downwelling acts to push the subsurface isopycnals downward, hence the decrease of density and freshening in the pycnocline with the positive NPGO (fig. 11b). The reverse processes may occur for the negative NPGO phase: atmospheric warm advection leads to less heat loss from ocean and retreat of sea ice cover, thus warming and freshening of surface water. The surface density decreases to a large extent. The cyclonic wind stress forcing and the subsequent upwelling induce the heaving of subsurface isopycnals, hence increased density and salinification in the pycnocline. Once the pycnocline water property changes, it gets less exposure to surface flux damping and tends to persist for more than one year. This may at least partly account for the dramatic decadal oscillation of Bering SST and sea ice extent anomalies after the year of 2007.

To further verify the sensitivity of SBII to the NPGO-oriented physical processes, the spring wind stress and Ekman pumping rate anomalies are composited based on the SBII high minus low index years as shown in fig. 12. The spring Bering sea ice cover expansion corresponds to the prevailing northerly over the Bering Sea and a large-scale anticyclonic circulation and upwelling in the North Pacific Ocean in the early period, which resembles the PDO-related pattern (fig. 8a). For the later period, there are anomalous anticyclonic wind stress curl over the Bering Sea and downwelling in association with the increased ice extent, which resembles the NPGO-related pattern (fig. 8d). In addition, the Bering Basin water property changes in association with the

SBII high minus low index years are also presented in fig. 13. In the early period, the ice expansion corresponds to the surface cooling but little change in the salinity and density. In the later period, the ice expansion corresponds to much larger magnitude of surface cooling as well as salinity and density anomalies. The change can also extend to pycnocline layer with the reverse of salinity and density anomalies vertically, which almost reproduces the NPGO-related T/S/Rho anomalies in fig. 11b.

In conclusion, it is the NPGO change and its synchronization with the PDO that trigger the recent decadal oscillation of Bering sea ice extent. This notable decadal oscillation in the Bering sea plays an important role to interpret recent total Arctic sea ice extent change. In the background of global warming and polar amplification, the total Arctic ice volume and thickness decrease persistently. It is conceivable that the future change of Arctic sea ice extent is more sensitive to the complicated air-sea coupling processes. Therefore, to better understand and predict the future Arctic sea ice change, we should pay more attention to these large-scale coupled modes such as Pacific decadal variability and Atlantic multi-decadal oscillation.

Fig. 3 Composites of (a)&(b) sea surface temperature anomalies; (c)&(d) subsurface (165m) water temperature anomalies based on the SBII high minus low index during the prior-2007 period (left panels) and the post-2007 period (right panels). Gray rectangle defines the area of Bering Sea (170-200E, 55-65N). Solid (dashed) lines enclose the positive (negative) values that are significant at the 95% confidence level.

Fig. 4 (a) Regression of March-May mean sea surface temperature anomalies (color shading) and sea level pressure anomalies (contour) on the PDO index during the period of 1979-2006. The contour interval is 0.2hPa. (b) Same as (a), but regression on the NPGO index. (c) & (d), same as (a) & (b), but during the period of 2007-2018.

Fig. 5 241 months running regression of the HadEN4.2.1 g10 Bering sea temperature anomalies on the PDO index (a) and on the NPGO index (b). All the data are preprocessed by 13 months low-pass filter. Solid (dashed) lines enclose the positive

(negative) regression coefficients that are significant at the 95% confidence level. (c) 241 months running correlation between the PDO and NPGO indices. Black and red lines indicate the original monthly correlation and 13 months low-pass filtered correlation respectively.

Fig. 6 (a) Time series of the annual mean PDO index (black bar) superposed by its decadal component (blue line) during the period of 1950-2018. The red dashed lines indicate the critical year of 2007. (b) Same as (a), but the NPGO index.

Fig. 7 (a) & (b) Regression of March-May mean surface turbulent (sensible + latent) heat flux onto the PDO (left panels) and NPGO (right panels) indices during the period of 1979-2006. (c) & (d) Same as (a) & (b), but for the later period. Solid (dashed) lines enclose the positive (negative) regression coefficients that are significant at the 95% confidence level.

Fig. 8 Same as Fig. 7, but for the regression of surface wind stress (vector) and Ekman pumping rate (color shading) anomalies. Solid (dashed) lines enclose the positive (negative) regression coefficients of Ekman pumping that are significant at the 95% confidence level. The black (gray) vectors indicate the wind stress regression coefficients are significant (insignificant) at the 95% confidence level.

Fig. 9 (a) Regressions of surface (5m) density (color shading) and zonal and meridional velocities (vector) onto the PDO index. Solid (dashed) lines enclose the positive (negative) density anomalies that are significant at the 95% confidence level. Only the velocity anomalies that are significant at the 95% confidence level are plotted. (b) Same as (a), but for the subsurface layer (165m).

Fig. 10 Climatological vertical profile of spring (March-May) Bering deep basin (160-190E, 50-60N) mean temperature and salinity (a) and density (b).

Fig. 11 Regression of spring (March-May mean) Bering deep basin temperature, salinity and density on the NPGO index during the 1979-2006 period (a) and the 2007-2017

period (b) for each depth layer. The marks of black dot, red cross and blue circle denote the regression coefficients that are significant at the 85% significant t test for density, temperature and salinity respectively.

Fig. 12 Composites of spring wind stress (vector) and Ekman pumping rate (color shading) based on the SBII high minus low index during the period of 1979-2006 (a) and the period of 2007-2018 (b).

Fig. 13 Same as fig. 11, but the composites based on SBII high minus low index.

Some specific comments:

1. Page 1, line 19: The decrease of pan-Arctic ice extent in recent decades has not really been "continuous", as there have been ups and downs associated with interannual variability. One can also question the statement in line 20 that the "rate of ice loss accelerates from the 1990s". If one calculates the trend of pan-Arctic September ice extent for the 2007-2018 period, the trend is essentially zero.

Reply: Some expressions are indeed not precise. We corrected them in the new manuscript. Thank you for pointing out this! The revision made in Page 1, line 24: Arctic sea ice has experienced a decline tendency and line 25: It is noteworthy that the remarkable ice retreat from the late 1990s.

2. Page 1, line 32: Not a sentence.

Reply: revised.

3. Page 4, line 32-33: In another example of the dated nature of the study, the PDO has been positive for nearly the entire period since 2015.

Reply: Thank you for reminding us about this! We extend our analysis period and reveal close linkage between the phase change of PDO and NPGO and the decadal oscillation of Bering Sea ice. Details please see the reply to your main point 2.

4. Page 6, lines 1-4: Figure 7 needs some elaboration, including an explanation of how

Ekman pumping contributes to the proposed linkage between the atmosphere and the Bering Sea water temperatures and sea ice. In particular, what happens dynamically when Ekman pumping changes from negative to positive? What causes the sign reversal, and how do upwelling and downwelling relate to changes in water temperature? In other words, please present more information on the mechanistic linkage alluded to here.

Reply: We agree with your viewpoint and explore the detailed oceanic physical response to the atmospheric forcing in the revised manuscript. Details please see the reply to your main points.

5. Page 6, lines 22-31: There should be some description of the NPO and NPO-NPGO, and some supporting references. As it now stands, the NPO and NPGO seem to appear out of the blue.

Reply: Thank you so much for providing these clues! According to your suggestions, we compared the roles played by the PDO and NPGO in the Bering sea ice decadal changes and relevant physical processes. The conclusion is made that the NPGO changes and its synchronization with PDO mode may trigger the decadal change of Bering Sea. Details please see the reply to your main points.

[Figure]

**Fig. 1.** Decadal change of total Arctic sea ice extent and its standard deviation

[Figure]

**Fig. 2.** Normalized ChukberSIE index in summer and spring

[Figure]

**Fig. 3.** Composites of SST and 165m T based on SBII index for the two periods

[Figure]

**Fig. 4.** Regressions of SST and SLP on the PDO and NPGO index for the two periods

[Figure]

**Fig. 5.** Running regression of Bering sea temperature on the PDO and NPGOindex and running
correlation between the two indices

[Figure]

**Fig. 6.** Annual mean PDO and NPGO indices

[Figure]

**Fig. 7.** Regression of surface turbulent heat flux onto the PDO and NPGO indices for the two periods

[Figure]

[Figure]

**Fig. 8.** Regressions of surface wind stress and Ekman puming rate onto the PDO and NPGO for the two periods

[Figure]

**Fig. 9.** Regressions of surface and subsurface density and zonal and meridional velocities onto the PDO

Interactive
comment

[Figure]

**Fig. 10.** Climatological vertical profile of spring Bering basin mean water properties

[Figure]

**Fig. 11.** Regression of spring Bering basin temperature, salinity and density onto the NPGO for the two periods

[Figure]

**Fig. 12.** Composites of spring wind stress and Ekman pumping rate based on the SBII index

[Figure]

**Fig. 13.** Composites of Bering basin temperature, salinity and density based on the SBII index for the two periods

---

## Author Response (AR1)

**Review #1 by John E. Walsh**

This paper shows that the Bering Sea was a major contributor to the increased amplitude of the seasonal cycle of pan-Arctic ice extent in the first decade of the 21st century. Perhaps more importantly, it addresses a topic that is in need of further understanding: the variability of Bering Sea ice extent over decadal timescales. While the paper contains some interesting results, it comes up short in two respects: (1) recent events have passed it by, and (2) the explanation of the Bering's decadal variations is largely speculative.

With regard to (1), the paper's analysis ends in 2015. Since 2015, the Bering Sea has been the region of perhaps the most remarkable extremes of sea ice extent in the historical record: the extreme minima of 2018 and 2019. Bering ice extents during these two winters were only about 50% of the previous minima of the satellite era. This represents a spectacular reversal from the positive anomalies of 2008-2013 that shape the results and conclusions of the present paper. The recent extreme minima have put the Bering at the forefront of climate discussions. There are obvious implications for the Bering Sea ice trends highlighted in Figure 1 and elsewhere in the paper. As a result, the paper is seriously dated before it even gets into the publication process.

Reply: Thank you for reminding us of this important information! We followed your advice and updated all the available datasets to present: (1) NSIDC monthly sea ice concentration data from November 1978 to June 2019; (2) ECMWF ERA-interim meteorological fields data from January 1979 to April 2019; (3) ECMWF ORA-S4 ocean fields data from January 1958 to December 2017; (4) JISAO PDO index from January 1900 to September 2018; (5) NCEP GODAS ocean assimilation data from January 1980 to May 2019; (6) HadEN4.2.1 g10 ocean subsurface analyses data from January 1950 to December 2018; (7) WHOI OAflux data from January 1958 to December 2018. To probe into the physical mechanism related to the sea ice change, we also downloaded the NPGO index constructed by Di Lorenzo from January 1950 to December 2018. All above changes are described in the revised paper Section 2.

      Once total Arctic sea ice extent time series is extended to the latest, its seasonal variation magnitude seems drop down again. That is, 2013-2018 annual standard deviation is overall lower than that of 2007-2012, representing a flavor of decadal scale oscillation feature (Fig. 1a). Fig. 1b shows the standard deviation differences for each month. It is obvious that discrepancies between the three periods (2007-2012 minus 1979-2006, and 2013-2018 minus 2007-2012) are primarily come from the summer (Aug-Oct, peak in Sep) and spring (Mar-May, peak in April) seasons. The spatial patterns further show that the Pacific section including the Chukchi sea and the Bering Sea (gray fan) as well as the Laptev and eastern Siberian seas is the major contributor to these decadal changes of total

Arctic sea ice extent (Fig. 1c & d). The normalized Chukchi-Bering SIE indices in ASO and MAM (Fig. 2) both exhibit abrupt change since 2007. In summer this change can be viewed as a mean state mutation or regime shift (Fig. 2a), while in spring it is more like a decadal oscillation (Fig. 2b). It seems that the annual cycle of total Arctic sea ice extent increases because of the persistent ASO Chukchi-Bering SIE retreat over the 2007-2018, while the increased decadal variability in MAM contributed to a strengthening of the seasonal cycle 2007-2012 relative to 2013-2018.

We made these revisions in the section 3.1 of the revised manuscript.

(2) The explanation for the Bering Sea ice history, especially its decadal-scale variability, leaves the reader wondering about the value added. The authors show that there has been a cooling of the Bering Sea at depth, coincident with the 2008-2013 positive sea ice anomalies, and that the water temperatures generally correlate well with the Bering ice extent over the entire record. But the reasons for the water temperature variations are only discussed speculatively: there is mention of Bering Strait throughflow (p6, lines 12-21) and atmospheric circulation variations (p5 bottom - p6 top), but there is no rigorous analysis. The text does not even indicate whether there was an increase in the heat exported through Bering Strait in the 2007-2008 timeframe, nor whether there would be a compensating inflow of heat from the south. The hypothesis about changes in ocean currents and Ekman pumping is also not supported by hard evidence, as no ocean current data were included in the analysis. The linkage between ice extent and water temperature does not distinguish cause and effect, and even the possible linkage with the PDO is rather murky in view of the lead-lag correlation in Figure 3d. In short, the paper does not provide a compelling explanation for the Bering Sea winter/spring ice expansion of 2008-2013 (let alone the spectacular abrupt decrease in 2018 and 2019).

Reply: Thank you for your suggestions! As we noted in the original manuscript, the spring Bering Sea ice high index is associated a SLP dipole pattern prior-2007 but an anomalous high pressure thereafter. Furthermore, the anomalous Bering SST anomalies exhibits a close connection with North Pacific large-scale SST pattern in the recent decade, in great contrast to the isolated Bering SST variability before 2007. In addition, the SST anomalies extend down to the subsurface after 2007. This remarkable pattern change suggests that the spring Bering Sea ice variability is getting predominated by the ocean dynamical processes in Pacific.

The extra-tropical North Pacific climate variability is dominated by two decadal-scale ocean-atmosphere modes, the Pacific Decadal Oscillation (PDO) and the North Pacific Gyre Oscillation (NPGO). Further analyses reveals that the NPGO pattern exhibits great decadal change and becomes phase-lock with PDO in the post-2007 period. The effects of PDO and NPGO on the Bering Sea temperature anomalies as well as the associated dynamical processes were

explored. It is the NPGO change and its synchronization with the PDO that trigger the recent decadal oscillation of Bering Sea ice extent. This notable decadal oscillation in the Bering sea plays an important role to interpret recent total Arctic sea ice extent change. In the background of global warming and polar amplification, the total Arctic ice volume and thickness decrease persistently. It is conceivable that the future change of Arctic sea ice extent is more sensitive to the complicated air-sea coupling processes. Therefore, to better understand and predict the future Arctic sea ice change, we should pay more attention to these large-scale coupled modes such as Pacific decadal variability and Atlantic multi-decadal oscillation.

The detailed physical mechanisms that account for the decadal oscillation of spring Bering Sea ice extent is presented in the revised manuscript section 3.2-3.4.

Some specific comments:
1.    Page 1, line 19: The decrease of pan-Arctic ice extent in recent decades has not really been "continuous", as there have been ups and downs associated with interannual variability. One can also question the statement in line 20 that the "rate of ice loss accelerates from the 1990s". If one calculates the trend of pan-Arctic September ice extent for the 2007-2018 period, the trend is essentially zero.
Reply: Some expressions were imprecise. We corrected them in the new manuscript. Thank you for pointing this out! The revision made in Page 1, line 24-25: "Arctic sea ice has declined along with the Arctic amplification of global warming. It is noteworthy that the remarkable ice retreat from the late 1990s".

2.    Page 1, line 32: Not a sentence.
Reply: revised.

3.    Page 4, line 32-33: In another example of the dated nature of the study, the PDO has been positive for nearly the entire period since 2015.
Reply: Thank you for reminding us about this! We extend our analysis period and reveal close linkage between the phase change of PDO and NPGO and the decadal oscillation of Bering Sea ice. Details please see the reply to your main point 2.

4.    Page 6, lines 1-4: Figure 7 needs some elaboration, including an explanation of how Ekman pumping contributes to the proposed linkage between the atmosphere and the Bering Sea water temperatures and sea ice. In particular, what happens dynamically when Ekman pumping changes from negative to positive? What causes the sign reversal, and how do upwelling and downwelling relate to changes in water temperature? In other words, please present more information on the mechanistic linkage alluded to here.
Reply: We agree with your viewpoint and explore the detailed oceanic physical response to the atmospheric forcing in the revised manuscript. Details please see

the reply to your main points.

5. Page 6, lines 22-31: There should be some description of the NPO and NPO-NPGO, and some supporting references. As it now stands, the NPO and NPGO seem to appear out of the blue.
Reply: Thank you so much for providing these clues! According to your suggestions, we compared the roles played by the PDO and NPGO in the Bering sea ice decadal changes and relevant physical processes. The conclusion is made that the NPGO changes and its synchronization with PDO mode may trigger the decadal change of Bering Sea. Details please see the reply to your main points.

**Review #2**

This paper presents an analysis of Bering and Chukchi Sea ice, showing contrasting decadal variability with decreases in the summer-autumn and increases in the winter-spring. Sea ice, atmospheric reanalysis, and ocean temperatures and velocities are examined to explain this behavior. The authors find that the pattern is likely due to subsurface cooling in the northern Bering Sea.

General comment

Overall, this is well-written paper and the analysis is competent. However, I have a few notable issues, listed below. The second issue in particular is critical and is the major reason for the decision of Major Revision.

1. The authors generally treat the Bering and the Chukchi Seas as one system, but I'm not sure if this really appropriate. For example, they say that the B-C shows decreases in the summer-autumn and increases in the winter-spring. However, the Bering is ice-free during summer-autumn, so the decrease is in the Chukchi, not really the B-C. Similarly, the Chukchi is completely ice-covered during winter-spring, so the increase there is really in the Bering, not the B-C.

During the analysis, they do start to treat the two more separately, but their approach is generally that these are two closely inter-connected systems and I'm not convinced that this is the case. There is indeed connection through the Bering Strait, and though that is narrow and shallow, there has been much research demonstrating that inflow of warm Pacific waters has contributed to summer ice loss in the Chukchi. But the Chukchi is also connected to the rest of the Arctic Ocean, particularly the neighboring seas — the Beaufort and the East Siberian. For example, the Beaufort Gyre circulation transports ice into the Chukchi and much of that ice is thicker, multi-year ice. This aspect seems to be ignored in the

paper- particularly the role of changes in multi-year ice, which has been rather dramatic in recent years.

 And while the Bering influences the Chukchi, I don't think the reverse is generally true – the ocean waters flow into the Chukchi. There can be some sea ice outflow, but this is pretty rare and ephemeral. So when talking about the B-C, it's really more of a one-way street, not an interacting system. The main connection would be through the atmosphere where circulation patterns can affect the transport of heat, etc. between the two regions.

 Maybe this is more semantics on my part. In the body of the paper, the authors seem to be clearer and focus more on the Bering Sea and then interaction with the Arctic Ocean in general. So maybe some changes in phrasing in places would help here.

 Reply: We very much agree that the Chukchi Sea ice and Bering Sea ice vary in different seasons and under the different mechanisms. We apply the term of ChukBerSIE in order to stress that the abrupt change of total Arctic sea ice extent variance is originated from this Pacific sector, rather than other Arctic areas. We recognize that the summer Chukchi sea ice variability is closely connected to the inner Arctic Ocean processes, but the spring Bering Sea ice is more under the spell of local climate variability. While much attention is paid to the drastic summer Arctic ice cover reduction, few people care about the spring Bering sea ice variability. Our preliminary analysis reveals that the Bering Sea ice has experienced a strengthening decadal oscillation, which at least partly accounts for the abrupt change of total Arctic sea ice variance in the recent decade. However, we did not know the detailed processes that trigger this decadal oscillation of Bering Sea ice. This is the motivation of our study.
    We greatly appreciate your suggestion about the phrasing change, and we do some revision to specify the difference between the Chukchi and Bering regions in Page 4 Line 26-27: "though the sea ice variations in the Chukchi Sea and the Bering Sea are probably subject to different dynamical processes and thermal conditions". And we further show and compare the difference between the summer Chukchi sea ice and spring Bering Sea ice in Page 4 Line 28- Page 5 Line 4 (Fig. 2).

 2. The other main issue is, unfortunately, the timing of this paper. As the authors are probably aware, the situation in the Bering has completely reversed over the last two winters. Both 2017-2018 and 2018-2019 were record or near-record lows over most of the winter-spring. I recognize that things can change and that papers have to be published at some point and new data will come in after submission. However, this paper stops with 2015, so after the last two winter-springs, it feels particularly out date... Even without updated ocean data, at least some of the analysis could be updated.

Because the change is so substantial, I feel the authors should really redo the analysis through at least 2018. As noted above, I recognize that at some point, analysis has to stop and a manuscript must be prepared and submitted. But this paper just feels too out of date to be particularly relevant. At the very least, if the authors can't update their analysis, they should thoroughly discuss the recent changes in the Bering Sea ice cover and the implications for their study.

Reply: Thank you! We follow your suggestion to extend all the datasets to the latest time except for the ORA-S4. This dataset ends up in dec2017. Please refer to the reply to review 1 for the detailed data description. We then redo the analyses using the new datasets and make much more revisions on our conclusions and physical interpretation.

3. Overall, the analysis is rather cursory. The authors pull together the right datasets and the analysis that they do is reasonable. But it doesn't seem enough to really substantiate their hypotheses – particularly in light of #2 above. The paper is <6 pages long and while I enjoy short, concise papers, the analyses are not very deep and they don't dig into the data enough to definitively support their conclusions.

Reply: Thank you for your insightful comments! In the revised manuscript, we explore in detail the physical factors that trigger the recent Bering Sea ice change. We finally conclude that the NPGO mode change and its synchronization with the PDO in the recent decade are the principle cause for the decadal oscillation of Bering Sea ice. Detailed statistical and physical analyses are now included in the paper.

4. The final issue is several grammar errors—not major, but occasionally throughout the manuscript. Some are note below in Minor Comments, but a thorough proofreading for English grammar would be helpful.

Reply: Thank you for your careful check-up! The revised manuscript is thoroughly proofread for English grammar.

Specific comments

5. 16-17: "significant" and "insignificant" are each used. Do these refer to statistical significance? If so, the significance level should be specified. If not, then I would suggest different wording, e. g., substantial, notable, etc. to avoid confusion over the use of "significant".
Reply: Thank you! Yes, these refer to statistical significance with the confidence level of 95%. Revised.

Figure 1b: I find this figure a bit confusing. The 120W-60W is not labeled—that's the Canadian Archipelago. I recognize this isn't terribly relevant to the study, but odd to have just that longitude range unlabeled. More relevant though is: where is the Laptev? I think it's included in what is labeled "East Siberian" –if so, that should be noted. Maybe what would help is a map that has the seas labeled and the longitude boundaries outlined.

Reply: We make much revision on all the figures. In the new version, we emphasize the decadal differences of Arctic sea ice variance. And we outlined the Chukchi-Bering section as the key region of this decadal change. Details can be found in the reply to review 1.

Minor Comments

1, 9: "in recent decade"—a minor grammar issue, but it makes things ambiguous: is it "in the recent decade", or is it "in recent decades"? This is repeated elsewhere in the manuscript as well.

Reply: It should be "in the recent decade". Corrected. Thank you!

1, 27: "transit" should be "transition"

Reply: revised in Page2 Line 1, thank you!

2, 10: "urgent" not "urgently"

Reply: revised in Page 2 Line 21.

5, 22: "decline" not "descent"

Reply: revised.

6, 15: as noted above, I don't think you can any longer say "lengthening sea ice seasons" for the Bering—certainly not for the last two years.

Reply: Thank you! Revised in Page 11 Line 28- Page 12 Line 1: "Echoing to this increasing heat flux… lead to the recent Arctic climate change".

6, 29: In light of the last two years' worth of data, I think the question of whether the "amplifying seasonal cycle of Arctic sea ice cover will be sustained" has been answered, at least in terms of the Bering Sea: no!

Reply: Thank you! We revised it. It is not "amplifying" but "decadal oscillation".

[revised manuscript text omitted]

---

## Referee Report (RR1)

**Comments on revision of Yang et al.'s "The Arctic sea ice extent change connected to Pacific decadal variability"**

The authors have generally responded well to the comments of the two original reviews, which were quite consistent in their assessment of the paper's weaknesses. Most notably, the study period has been extended through 2018 to capture the recent precipitous decline of sea ice in the Bering Sea. This extension has required that the variations be cast into a framework of decadal variability, as reflected in the paper's new title. The framework of decadal (or at least multiyear) variability seems interesting and appropriate for the past 20 years, although it is not an outstanding feature of the pre-2000 portion of the sea ice time series (see Figure 2).

A second major revision is the addition is the authors' attempt to address mechanisms, which they do largely by relying on the NPGO and its phasing with the PDO. In particular, the authors argue that the character of the NPGO has changed in recent decades, and this change is tied to the emergence of the decadal (multiyear) character of the sea ice variations. While the authors provide some diagnostics to support the change in the NPGO and its manifestations, the analysis and interpretation still seem somewhat tenuous. The authors even note that the reasons for the change in the NPGO "deserves further investigation that is beyond the scope of this paper (p. 7, bottom). The proposed linkage will certainly not be the final word on the driving of the Bering Sea's sea ice trajectory, but I do give the authors credit for putting forth a linkage that can improved upon (or disproven) by future studies.

One lingering item for clarification is the strength of the NPGO signal. As I understand from the text (p. 6, lines 15-20) the NPGO is defined as an EOF of Pacific basin-scale sea surface height anomalies. If so, what portion of the total variance does it explain? Alternatively, how much atmospheric variance do the NPGO's atmospheric manifestations explain?

Lastly, given the paper's increased emphasis on multiyear variations that fall under the umbrella of internal variability, it would be appropriate to refer to the work on this subject by Rong Zhang and her GFDL collaborators. Examples include

Zhang, R. 2015: Mechanisms for low frequency variability of summer Arctic sea ice extent, PNAS, 112,doi: 10.1073/pnas.1422296112.

Lee, H C., T. L Delworth, A.Rosati, R.Zhang, W.G. Anderson, F. Zeng, C. A Stock, A. Gnanadesikan, K. W Dixon, and S.M Griffies, 2013: Impact of climate warming on upper layer of the Bering Sea. Climate Dynamics, 40, DOI:10.1007/s00382-012-1301-8.

Zhang, R., and T.R. Knutson, 2013: The role of global climate change in the extreme low summer Arctic sea ice extent in 2012 [in "Explaining Extreme Events of 2012 from a Climate Perspective"]. Bull. Amer. Meteor. Soc., 94 (9).

As a final comment, the English has been improved in the revision but there are still instances of faulty or awkward English. Examples: p. 5, line 2; p. 9, lines 12-12; p. 10, line 32.

---

## Author Response (AR2)

**Review #1 by Jonh E. Walsh**

Comments on revision of Yang et al.'s "The Arctic sea ice extent change connected to Pacific decadal variability"

The authors have generally responded well to the comments……., although it is not an outstanding feature of the pre-2000 portion of the sea ice time series (see Figure 2).

A second major revision is……, but I do give the authors credit for putting forth a linkage that can improve upon (or disproven) by future studies.

One lingering item for clarification is the strength of the NPGO signal. As I understand from the text (p.6, lines 15-20) the NPGO is defined as an EOF of Pacific basin-scale sea surface height anomalies. If so, what portion of the total variance does it explain? Alternatively, how much atmospheric variance do the NPGO's atmospheric manifestations explain?

Reply: Thank you for your careful consideration! The NPGO is defined as the $2^{nd}$ dominant mode of sea surface height variability ($2^{nd}$ EOF SSH) in the Northeast Pacific. It essentially reflects changes in the intensity of subtropical and subpolar gyre circulation. From our EOF analysis (Figure S7 in the manuscript), the explain variance of $2^{nd}$ EOF of SSH is ~14% in

the 1980s and 1990s and decreases to 10% in this century. More importantly, the spatial pattern of $2^{nd}$ EOF of SSH changes a lot with time. The monopole pattern in the later period (shown in Figure S7d) looks quite different from the original NPGO definition. In addition, the NPGO is thought to be associated with its atmospheric counterpart — the North Pacific Oscillation (NPO). From our analysis, the NPGO-NPO connection weakened in the recent decade and the atmospheric mode corresponding to the NPGO exhibits a quadrupole pattern (Figure 6d in the manuscript). Both the oceanic and atmospheric representations of NPGO undergo a strong change in these years, which may be related to its synchronization with the PDO mode. But the causality and physical processes for this change are unclear and need to further research.

Lastly, given the paper's increased emphasis on…… Examples include……

Reply: Thanks for recommendation! We referred to the relative papers (Section 4 the last paragraph).

As a final comment, the English has been improved in the revision but there are still instances of faulty or awkward English. Examples: p.5, line 2; p. 9, lines 12-13; p. 10, line 32.

Reply: Revised accordingly.

(1) P5: "collaborate to" → "result in"

(2) P9: "There is a sharp rising of temperature from ……below the permanent pycnocline." → "The relative warm subsurface water with its temperature exceeding 3.5℃ is sandwiched by the colder surface and deep water layers."

(3) P10:"The eastern Siberian is resided by……and southeast negative SLP anomalies (Fig. S3d)" → "An anomalous high pressure appears over the eastern Siberia. The "

**Review #2**

Review of "The Arctic sea ice extent change connected to Pacific decadal variability" by Yang et al.

General Comment: This paper is vastly improved over the initial submission…... I applaud the hard work of the authors in addressing my comments and improving the paper.

Specific Comments:

Intro, 1st sentence: "indicators of global climate change"

Intro, last paragraph, 1st sentence: use "rapid sea ice loss" instead of "fast

sea ice loss" to avoid confusion with the "fast sea ice" (fast ice) type.

Reply: corrected

Sec. 2, first paragraph: Just use "NASA", spelling out not needed. However, I would suggest spelling out what DMSP, SSMI, and SSMIS stand for. Also remove "team" after "NASA". What you mean (I think) is that the concentrations are derived using the "NASA Team" algorithm from the passive microwave radiances.

Reply: revised, thanks!

Sec. 2. On the sea ice data, did you use monthly gridded concentration fields and then calculate the monthly extents from those? That is okay, but can introduce a bias because the monthly gridded concentrations conflate temporal and spatial aspects. For example, a grid cell that is has 100% concentration for 5 days in a 30-day month exceeds the 15% threshold and will be considered "ice-covered" for the month and part of the extent. But that's very different than a cell that has 15% ice each day of the month. Ideally, when calculating monthly total sea ice extent, it is best to calculate individually for each day of the month and then average the daily total extents over the month. This is what is done for the NSIDC Sea Ice Index (https://nsidc.org/data/seaiceice_index/archives), which includes regional analyses for the Bering Sea and the Chukchi Sea (not

sure though if those regions are the same as what you use). Of course, for the spatial maps of concentration, you would use the monthly grids. As noted, for extent, there could be a bias. If it's possible to redo the analysis with the SII numbers – if they fit your spatial region criteria – that would be best. But at the least, I think noting how you calculated the extent – from the monthly average concentration maps.

Reply: We greatly appreciate your consideration! According to your suggestion, we repeat the ice extent calculation with daily sea ice concentration data and then average for each month. The new ChukBerSIE indices in summer and spring are shown in Fig. 1. The new indices almost reproduce the original ChukBerSIE indices (Figure 2 in the manuscript), with negligible differences. This demonstrates the robustness of our results. We follow your suggestion to note the calculation of sea ice extent in Section 2.

[Figure]

Fig.1 Normalized ChukBerSIE index in (a) summer (Aug-Oct mean); (b) spring (Mar-May mean). The daily sea ice extent is calculated from NSIDC daily sea ice concentration and then averaged for each month.

Sec. 3.1. It sounds like you used Nov. 1978 to Jun. 2019. This gives the maximum length, which is nice. But, with partial years at each end, the annual average trends are going to be skewed some. Probably not much, but for annual average it's better to include only full years (e.g., 1979-2018). For the seasonal, it is okay as long as any season is full. But you do have for example (Figure 1 and 2) the MAM period, which presumably would be 1979-2019, and the ASO period, which would be

1979-2018. I might suggest sticking with full calendar years for consistency throughout: 1979-2018. Another factor is that the 2019 is at the moment preliminary near-real-time data, which will be replaced with final fields. So, the 2019 values will change. This change is generally small and the NRT product is meant to be as consistent as possible as the final long-term record. Thus, I doubt it will affect the analysis. But if 2019 is used, then it should also be cited: http://nsidc.org/data/nsidc-0081.html.

Reply: Thanks for your suggestion! By calculating the annual averaged values and climatology, we excluded the incomplete data in 2019. The same did for most of seasonal mean analyses. Only in presenting the monthly total Arctic SIE time series (Figure 1a upper panel) and SBII (Figure 2b), we retained the latest data in 2019 in order to show the persistency of sea ice state both in the Bering Sea and throughout the whole Arctic region. As for the statistical and diagnostic analyses of ocean-ice-atmosphere interaction processes, the year of 2019 was not included at all. Thus it will not influence our results. We added the website you indicated in "data availability".

Sec. 3.1, paragraph starting "In view of…": "the eastern Siberian Sea" is mentioned. Is this the specific "East Siberian Sea" or the eastern part of some Siberian area? If it is the former, changed "eastern" to "East".

Reply: corrected.

Sec. 3.1, paragraph starting "Fig. 2 shows…": "collaborate to the" is awkward. Maybe "result in the"?

Reply: The expression was revised, thank you!

Sec. 3.3, 2nd paragraph: "deepening of the Aleutian Low" (add "the")

Reply: corrected.

Sec. 3.3, paragraph starting "Yeo et al. (2014)…": remove "until" before "after ~2007 when…"

Reply: corrected.

Sec. 3.4, paragraph starting "A NPGO-related…": "open water owing to the brine-rejection effect…" – need to add "the"

Reply: corrected.

[revised manuscript text omitted]

---

## Author Response (AR3)

In general, I am happy with your last round of replies. However, I find that the following issues still must be addressed:

1. The reviewer comment related to the strength of the NPGO signal must be addressed also in the paper, not only in the reply to the reviewer.

   Reply: Revised. Please see section 4, paragraph 3.

2. p.11, l.2: do you mean "over east Siberia" or "over the East Siberian Sea"?

   Reply: It is "east Siberia". Revised in page 8: line 28 and page 11: line 8, thank you!

3. Please have all your recent additions be checked by a native speaker, there are some minor grammar flaws in them.

   Reply: The manuscript has been thoroughly checked by the third author (a native English speaker). Thanks!

[revised manuscript text omitted]